# Gα13 restricts nutrient driven proliferation in mucosal germinal centers

Hang T. Nguyen[1], Moyi Li[1], Rahul Vadakath[1], Keirstin A. Henke[2], Tam C. Tran[3], Huifang Li[1], Maryam Yamadi[1], Sriranjani Darbha[1], Yandan Yang[1], Juraj Kabat[4], Anne R. Albright[1], Enoc Granados Centeno[1], James D. Phelan[1], Sandrine Roulland[1], Da Wei Huang[1], Michael C. Kelly[5], Ryan M. Young[1], Stefania Pittaluga[6], Simone Difilippantonio[2] & Jagan R. Muppidi[1]✉

Germinal centers (GCs) that form in mucosal sites are exposed to gut-derived factors that have the potential to influence homeostasis independent of antigen receptor-driven selective processes. The G-protein Gα13 confines B cells to the GC and limits the development of GC-derived lymphoma. We discovered that Gα13-deficiency fuels the GC reaction via increased mTORC1 signaling and Myc protein expression specifically in the mesenteric lymph node (mLN). The competitive advantage of Gα13-deficient GC B cells (GCBs) in mLN was not dependent on T cell help or gut microbiota. Instead, Gα13-deficient GCBs were selectively dependent on dietary nutrients likely due to greater access to gut lymphatics. Specifically, we found that diet-derived glutamine supported proliferation and Myc expression in Gα13-deficient GCBs in the mLN. Thus, GC confinement limits the effects of dietary glutamine on GC dynamics in mucosal tissues. Gα13 pathway mutations coopt these processes to promote the gut tropism of aggressive lymphoma.

GCs arise within B cell follicles in lymphoid tissue during immune responses and support the generation of high affinity antibodies[1]. B cells in GCs iteratively cycle between the light zone (LZ) and the dark zone (DZ). A small fraction of LZ cells that are positively selected by T cells show increased mechanistic target of rapamycin (mTOR) complex I (mTORC1) activity and expression of the proto-oncogene MYC before entering the DZ[2-4]. mTORC1 is a critical driver of cell growth that integrates a range of inputs including growth factors and nutrient availability[5], whereas MYC is a master regulator of cell growth and division whose transient expression in the LZ is required to sustain the GC reaction[2,3].

GCs are most often studied in the context of immunization or infection; however, GCs also form during homeostasis in mucosal tissues such as mLNs and Peyer's patches (PPs). These chronic GCs are thought to arise in response to stimuli derived from gut microbiota and diet and can show altered molecular dependencies compared to their immunized counterparts[6-10]. Whether specific dietary factors influence selection events in mucosal GCs is not clearly defined.

B cells that enter the GC can acquire deleterious mutations arising from aberrant somatic hypermutation that can cause lymphoma[11]. Diffuse large B cell lymphoma (DLBCL) is the most common form of human lymphoma and is characterized by substantial genetic heterogeneity. To make sense of this heterogeneity, subtypes of DLBCL have been defined by differential gene expression or specific genetic alterations[12,13]. These subtypes are derived from distinct cells of origin with distinct pathobiology. For instance, the GCB-like subtype (GCB-DLBCL) is derived directly from GCBs[11]. The most clinically aggressive subset of GCB-DLBCL is enriched for gene signatures with high MYC activity[13]. Loss of function mutations in *GNA13*, encoding the G-protein Gα13, are highly enriched in GCB-DLBCL with high MYC activity[13]. Additionally,

[1]Lymphoid Malignancies Branch, Center for Cancer Research, NCI NIH, Bethesda, MD, USA. [2]Gnotobiotics Facility, Frederick National Laboratory for Cancer Research, Leidos Biomedical Research, Frederick, MD, USA. [3]Precision Health Informatics Section, NHGRI NIH, Bethesda, MD, USA. [4]Research Technologies Branch, NIAID NIH, Bethesda, MD, USA. [5]Single Cell Analysis Facility, Center for Cancer Research, NCI NIH, Bethesda, MD, USA. [6]Laboratory of Pathology, Center for Cancer Research, NCI NIH, Bethesda, MD, USA. ✉e-mail: jagan.muppidi@nih.gov

loss of function *GNA13* mutations are enriched in Burkitt lymphoma (BL), which is defined by MYC translocations and commonly presents with mLN involvement[14–17]. The molecular basis of the association between Gα13 loss and increased MYC is unclear.

Gα13 signaling inhibits cellular migration and, in doing so, acts to confine GCBs to the niche at the center of the follicle[7,14]. In addition to its role in niche confinement, Gα13 signaling can suppress the accumulation of B cells in GCs[14]. It is thought that Gα13 signaling suppresses GC accumulation via inhibition of PI3K/Akt[11], but whether dysregulation of PI3K/Akt is the primary mechanism driving GC accumulation in the absence of Gα13 remains unclear.

In this study, we found that Gα13 suppressed DLBCL formation specifically in the mLN of mice. Gα13-deficient GCB expanded in mLNs but not in peripheral lymph nodes (pLNs). Unlike high Akt states that promoted LZ expansion and suppressed proliferation, Gα13 deficiency increased proliferation due to increased mTORC1 activity and Myc protein abundance in mLNs. We found that expansion of Gα13-deficient GCBs in mLNs was not dependent on gut microbiota or T cell help. Instead, dietary nutrients supported Gα13-deficient GCBs likely due to greater access to gut lymphatics. Finally, we found that dietary glutamine differentially supported Gα13-deficient mLN GCBs. Our data suggest that Gα13 suppresses lymphoma in mLNs by limiting the effects of dietary nutrients on GC-selective processes.

## Results

### Gα13 suppresses lymphoma in mesenteric lymph nodes

We assessed tumor incidence in cohorts of mice lacking Gα13 in mature B cells (*Cr2-cre Gna13*[f/f]) that were between 10 and 25 months of age. B cell-specific deletion of Gα13 resulted in spontaneous tumor formation in 27 of 34 mice with most animals developing tumors by 16 months of age (Fig. 1a,b). mLN involvement was observed in 25 of 27 tumor-bearing Gα13-deficient mice; in 12 mice, tumors were present in mLNs only (Fig. 1b). Mice heterozygous for *Gna13* in B cells also developed tumors, albeit at a lower frequency. We analyzed 11 Gα13-deficient mLN tumors histologically; 10 of 11 tumors showed expansion of sheets of large, atypical lymphocytes positive for the B cell marker B220, the GC marker GL7 lacking coincident staining of CD35[+] follicular dendritic cell (FDC) meshworks consistent with DLBCL (Fig. 1c and Extended Data Fig. 1). One of 11 tumors showed expansion of intermediate and large lymphocytes with FDC meshworks and follicle-like structures consistent with follicular lymphoma (Fig. 1c and Extended Data Fig. 1). Despite the highly penetrant development of GC-derived mLN tumors in aged Gα13-deficient animals, mLN GCBs were only modestly increased in 8-week-old Gα13-deficient animals (Fig. 1d and Extended Data Fig. 2a). Because selection in GCs occurs iteratively, a relatively small change in non-competitive environments can lead to large effects over time in competitive settings[18]. Therefore, we evaluated whether loss of Gα13 induced an advantage in a competitive environment. We analyzed mLN and immunized pLN GCs in bone marrow (BM) chimeras generated with a mixture of wild-type (WT) CD45.1/2 BM and CD45.2 BM that was WT (*Gna13*[f/+]) or Gα13-deficient (*Cr2-cre Gna13*[f/f]) (Fig. 1e). Loss of Gα13 led to a cell-intrinsic expansion

of GCBs in mLN GCs, consistent with our previous findings (Fig. 1f and Extended Data Fig. 2b)[14]; however, Gα13-deficiency did not result in a competitive advantage of GCBs in pLNs (Fig. 1g and Extended Data Fig. 2c). In PPs, Gα13 deficiency led to a smaller effect in comparison to mLNs (Extended Data Fig. 2d). Individual lobes of mLNs drain distinct segments of the small intestine and this compartmentalized drainage can lead to distinct immunological effects[19,20]. We found that ileal-draining lobes of mLNs showed the largest outgrowths of Gα13-deficient GCBs (Extended Data Fig. 2e). These data suggest that small intestine-derived cues support outgrowths of Gα13-deficient GCBs in mLNs and over time can promote the development of GC-derived tumors at this site.

To determine how loss of Gα13 affected homeostasis in established mLN GCs of non-irradiated mice we crossed *Gna13*[f/f] animals to animals carrying a GC-specific tamoxifen-inducible fate reporter allele[21] (*S1pr2-tdTomato; S1pr2-creETR2 Rosa26*[LSLtdTomato/+]) (Fig. 1h). In this system, administration of tamoxifen results in tdTomato labeling and loss of Gα13 in GCBs. In contrast to control animals, where labeled GCBs in mLNs steadily decrease over time after tamoxifen due to entry of unlabeled WT clones into the GC[22], labeled Gα13-deleted GCBs persisted in the mLN GC (Fig. 1h,i). Additionally, at 8 weeks following tamoxifen, unlabeled WT GCBs were suppressed in the presence of Gα13-deficient GCBs (Fig. 1j and Extended Data Fig. 2f). Notably, there was not an increased accumulation of labeled Gα13-deficient memory B cells in the mLNs or plasma cells in the BM at 8 weeks following tamoxifen, suggesting that, upon loss of Gα13, B cells are more likely to remain in the GC state (Fig. 1k and Extended Data Fig. 2g). We then asked whether Gα13 loss resulted in an increased mutational burden due to longer residence of Gα13-deficient clones in the mLN GC. We performed immunoglobulin heavy chain repertoire sequencing and found an increased frequency of mutations in variable regions in Gα13-deficient mLN GCBs, suggesting longer clonal residence in the GC (Fig. 1l). These data demonstrate that loss of Gα13 promotes a supercompetitor-like state leading to clonal longevity that likely contributes to the development of aggressive B cell lymphoma in mLNs.

### Gα13 suppresses mLN GC proliferation

One proposed pathway by which Gα13 suppresses tumorigenesis is inhibition of PI3K/Akt, but it is unclear whether this is the primary mechanism[11,14,23]. PI3K/Akt maintains GC polarity and high Akt activity is associated with accumulation of cells in the LZ[24,25]. To assess whether Gα13-deficiency primarily acts via inhibition of PI3K/Akt, we assessed how GC dynamics are perturbed in Gα13-deficiency versus settings with increased Akt activity. We reconstituted irradiated hosts with *Cr2-cre* BM transduced with retrovirus expressing myristoylated Akt (myr-Akt) downstream of a loxP–stop–loxP cassette. Expression of myr-Akt in mature B cells resulted in cell-intrinsic expansion of GCBs in the mLNs and to a lesser extent in immunized pLNs (Extended Data Fig. 3a–d)[6,7]. Myr-Akt expression resulted in LZ GCB expansion in mLNs and pLNs (Extended Data Fig. 3e,f). Consistent with expansion of the LZ and reduction of the DZ (where most proliferation occurs), myr-Akt

**Fig. 1 | Gα13 suppresses tumor development and GCB cell clonal persistence in the mLN. a**, Tumor-free survival of animals with B cell-specific Gα13-deficiency (*Gna13* KO; *Cr2-cre Gna13*[f/f]) or WT littermates aged up to 750 days. $n = 36$ WT, $n = 34$ *Gna13* KO. ****$P < 0.0001$ log-rank (Mantel–Cox) test. **b**, Anatomic location of tumors in aged *Gna13* WT (*Gna13*[f/+]), *Gna13* heterozygous (Het) (*Cr2-cre Gna13*[f/+]) or *Gna13* KO (*Cr2-cre Gna13*[f/f]) animals. Example gross image of mLNs from 16-month-old animals on left. Scale bar, 1 cm. **c**, Pathological classification of mLN tumors in aged Gα13-deficient animals. FL, follicular lymphoma. $n = 11$. Examples are shown in Extended Data Fig. 1. **d**, GCBs in mLNs of 8-week-old bred animals. Data are from four experiments, $n = 7$ littermates, $n = 4$ *Gna13* KO. *$P = 0.0357$ unpaired two-tailed Student's *t*-test. **e**, Experimental scheme for **f** and **g**. s.c., subcutaneous. **f,g**, Percentages of CD45.2 follicular B cells (FoBs)

and GCBs or the ratio of CD45.2 GCB to CD45.2 FoBs in mLNs (**f**) or pLNs (**g**) of mixed BM chimeras. Data are pooled from two experiments, $n = 12$ control, $n = 10$ *Gna13* KO. ****$P = 9.58 \times 10^{-12}$ and $1.41 \times 10^{-6}$, respectively in **f**, unpaired two-tailed Student's *t*-test. **h**, Experimental scheme and gating strategy for fate-mapped GCBs and memory B cells for **i–k**. **i–k**, tdTomato[+] GCs (**i**) or tdTomato[−] or tdTomato[+] mLN GCBs as a percentage of live cells (**j**) or tdTomato[+] memory B cells (B220[+]IgD[lo]CD38[high]Fas[int]GL7[−]; **k**) in *S1pr2-tdTomato* mice at 1, 5 or 8 weeks following tamoxifen administration. Data are from ten experiments, $n = 5, 7, 13, 10, 12$ and 12. ***$P = 0.0002$, ****$P = 1.77 \times 10^{-5}$ in **i**, **$P = 0.0034$, *$P = 0.0115$ in **j**, unpaired two-tailed Student's *t*-test. **l**, Mutation frequency per read in IgV$_H$ repertoire sequencing of mLN GCBs. $n = 4$. **$P = 0.0047$, *$P = 0.0252$, **$P = 0.0094$, *$P = 0.0223$, *$P = 0.025$ unpaired two-tailed Student's *t*-test. KO, knockout.

expressing GCBs showed reduced incorporation of bromodeoxy-uridine (BrdU) after in vivo labeling for 30 min (Fig. 2a and Extended Data Fig. 3g); however, in the mLNs of Gα13-deficient mixed chimeras,

loss of Gα13 did not alter polarity but did result in increased GCB proliferation (Fig. 2b and Extended Data Fig. 3h). In contrast to mLNs, Gα13-deficiency in pLNs resulted in an LZ bias and reduced proliferation

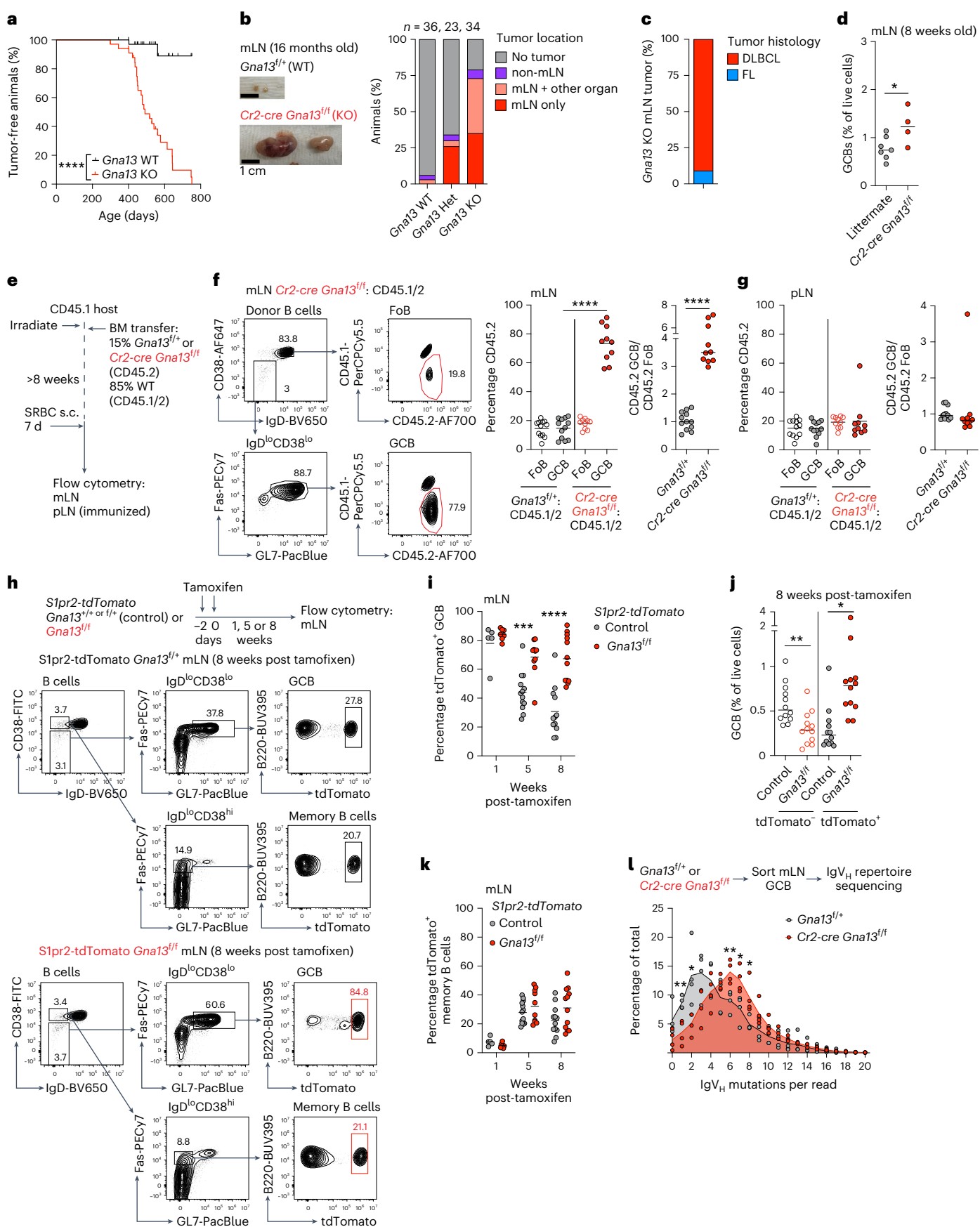

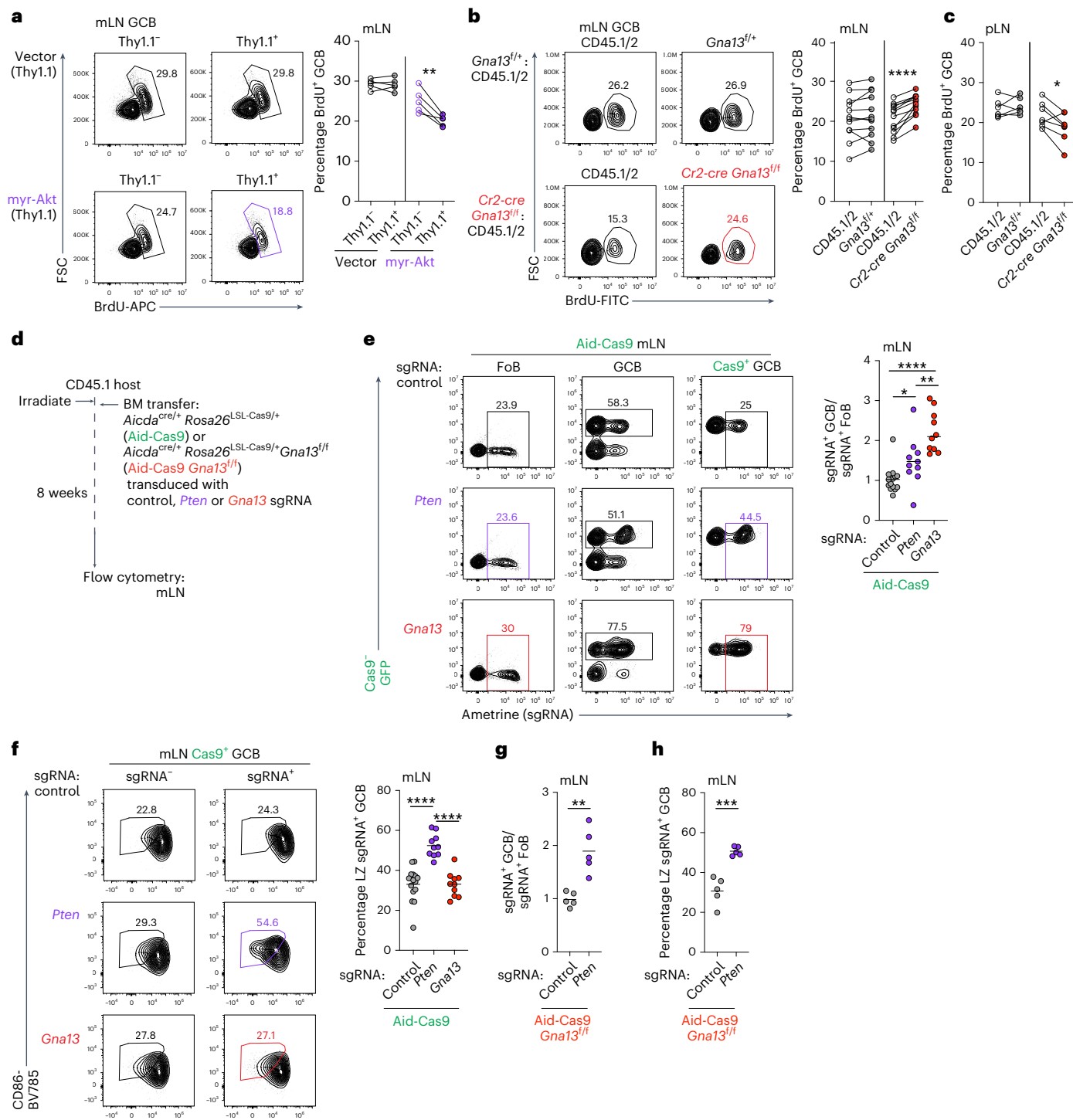

**Fig. 2 | Gα13 suppresses mLN GC B cell proliferation. a**, Percentages of proliferating (BrdU⁺) GCBs among untransduced (Thy1.1⁻) or transduced (Thy1.1⁺) cells in mLNs of myr-Akt chimeras generated as in Extended Data Fig. 3a. Data are from one experiment representative of two, $n$ = 5 mice per group. **$P$ = 0.0098 paired two-tailed Student's $t$-test. **b,c**, Percentages of proliferating (BrdU⁺) GCBs among WT (CD45.1/2) or WT (*Gna13*$^{f/+}$) or Gα13-deficient (*Cr2-cre Gna13*$^{f/f}$) cells in mLNs (**b**) or pLNs (**c**) of mixed BM chimeras. $n$ = 12 mice per group in **b** pooled from three experiments and $n$ = 6 and 7 mice per group pooled from two experiments in **c**. ****$P$ = 8.49 × 10⁻⁵ for **b**, *$P$ = 0.028 for **c** paired

two-tailed Student's $t$-test. **d**, Experimental scheme for **e–h. e,f**, Ratio of sgRNA-expressing Cas9⁺ GCB (sgRNA⁺ GCB) to sgRNA⁺ FoB (**e**) or percentage of LZ sgRNA⁺ GCB (**f**) in Aid-Cas9 BM chimeras targeting *Pten* or *Gna13*. Data are from three experiments with $n$ = 15, 10 and 10 mice per group. *$P$ = 0.024, **$P$ = 0.0084, ****$P$ = 2.77 × 10⁻⁷ unpaired two-tailed Student's $t$-test. **g,h**, Ratio of sgRNA-expressing Cas9⁺ GCBs (sgRNA⁺ GCBs) divided by sgRNA⁺ FoBs (**g**) or percentage of LZ sgRNA⁺ GCBs (**h**) in Aid-Cas9 *Gna13*$^{f/f}$ BM chimeras targeting *Pten*. Data are from one experiment, $n$ = 5. **$P$ = 0.0019 for **g** and ***$P$ = 0.0003 for **h**, unpaired two-tailed Student's $t$-test.

(Fig. 2c and Extended Data Fig. 3i). Additionally, consistent with the expanded LZ in pLNs, pAkt was increased in Gα13-deficient LZ GCBs in pLNs but not mLNs in mixed chimeras (Extended Data Fig. 3j–l).

These data suggest that cues in mLNs enable increased proliferation in the setting of Gα13 loss and that, in the absence of mLN-specific cues, Gα13-deficiency leads to increased Akt activity.

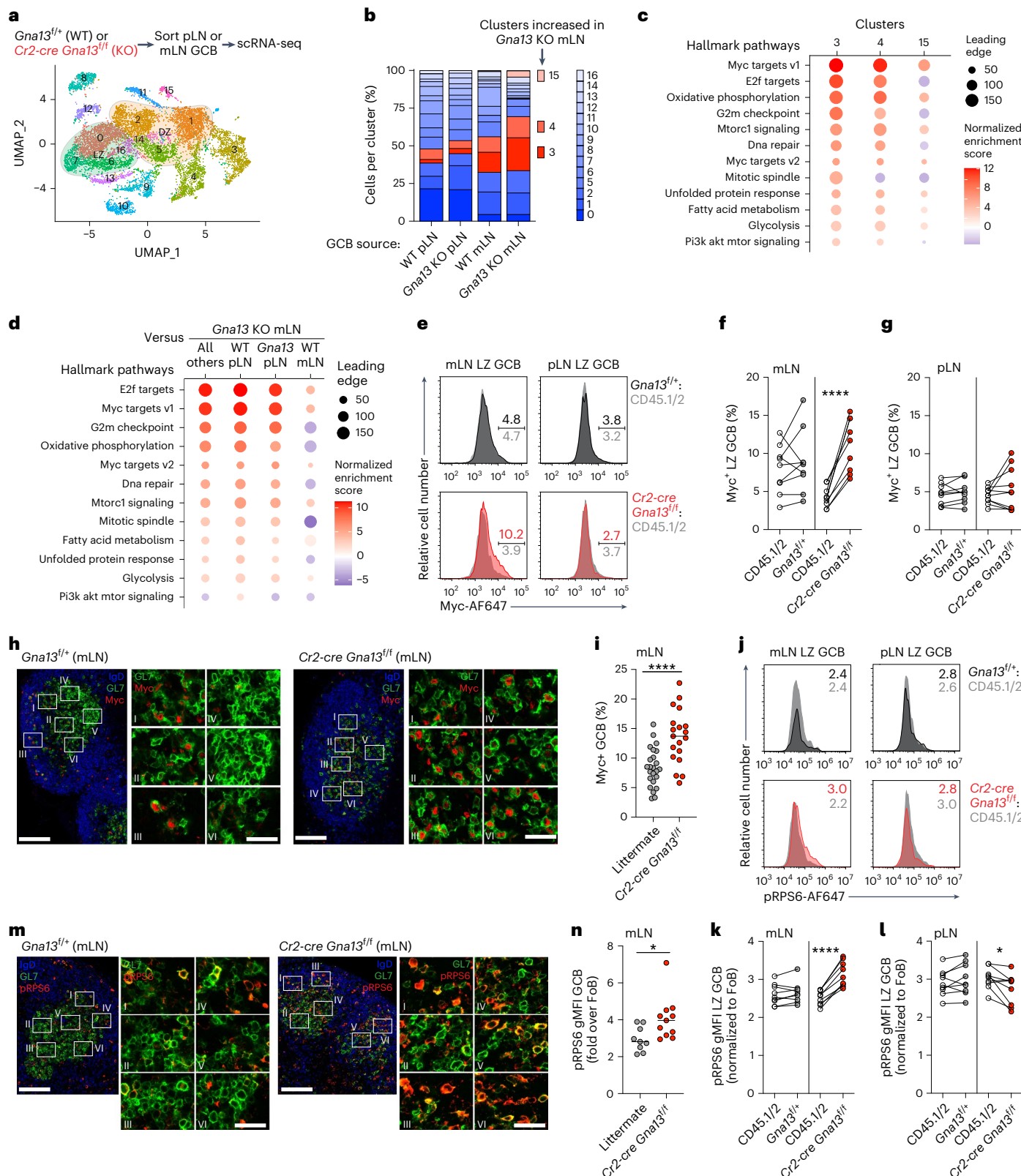

To determine whether a more physiological perturbation of PI3K/Akt could phenocopy Gα13-deficiency in mLNs, we reconstituted irradiated hosts with BM conditionally expressing Cas9 in GCBs (Aid-Cas9; *Aicda*^cre/+ *Rosa26*^LSLCas9/+) transduced with retrovirus expressing small guide RNA (sgRNA) targeting *Pten*, encoding a lipid phosphatase that counteracts PI3K or *Gna13* that also expressed the fluorescent reporter ametrine (Fig. 2d). Loss of Gα13 in GCBs resulted

in outgrowths of mLN GCBs but did not alter GC polarization (Fig. 2e,f). Loss of Pten, however, resulted in a comparatively smaller increase in GC outgrowth but was able to cause substantial bias toward the LZ (Fig. 2e,f). Finally, to assess whether the combined loss of Pten and Gα13 was epistatic or synergistic, we examined chimeras reconstituted with Aid-Cas9 *Gna13*^f/f BM expressing *Pten* sgRNA (Fig. 2d). In the absence of Gα13, loss of Pten was still able to promote GC outgrowths and LZ

**Fig. 3 | Gα13 suppresses mTorc1 signaling and Myc protein expression in mLN GCB cells. a**, Unsupervised clustering of mLN or pLN GCBs from WT or Gα13-deficient mice. Clusters with the highest LZ or DZ signatures are indicated. **b**, Fraction of cells in each cluster in WT or Gα13-deficient GCBs from pLN or mLN. **c**, Hallmark gene sets enriched in clusters with increased representation in Gα13-deficient mLNs. **d**, Gene sets enriched in Gα13-deficient mLNs compared to all other samples. **e–g**, Intracellular FACS of Myc in LZ cells from mLNs or pLNs (**e**) or percentage of LZ GCBs expressing Myc in mLNs (**f**) or pLNs (**g**) in mixed BM chimeras. Data are from two experiments, $n = 9$. ****$P = 7.93 \times 10^{-5}$ paired two-tailed Student's $t$-test for **f**. **h**, Confocal microscopy of mLN GCs stained for IgD, GL7 and Myc. Scale bars, 100 μm (whole GC) or 25 μm (inset). **i**, Percentage of Myc+ GCBs in mLNs quantified by histocytometry. Data are

from three experiments, $n = 4$ littermate, $n = 3$ Gna13 KO. Each symbol refers to the frequency of Myc+ cells in one GC. ****$P = 9.13 \times 10^{-5}$ unpaired two-tailed Student's $t$-test. **j–l**, Intracellular FACS of phospho-RPS6 S240/244 (pRPS6) in LZ cells from mLNs or pLNs (**j**) or gMFI of pRPS6 normalized to FoBs in LZ GCBs in mLNs (**k**) or pLNs (**l**) in mixed BM chimeras. Data are from nine experiments, $n = 9$. ****$P = 9.14 \times 10^{-6}$, *$P = 0.0207$ paired two-tailed Student's $t$-test. **m**, Confocal microscopy of mLN GCs stained for IgD, GL7 and pRPS6. Scale bars, 100 μm (whole GC) or 25 μm (inset). **n**, gMFI of pRPS6 in GCBs relative to FoBs in mLNs quantified by histocytometry. Data are from three experiments, $n = 6$ littermate, $n = 3$ Gna13 KO. Each symbol refers to the gMFI of pRPS6 in one GC. *$P = 0.022$ unpaired two-tailed Student's $t$-test. gMFI, geometric mean fluorescence intensity.

expansion (Fig. 2g,h). Collectively, these data suggest that expansion of Gα13-deficient cells in the mLN, the site at which tumors eventually form, is due to increased proliferation and is not primarily driven by dysregulated PI3K/Akt.

## Gα13 suppresses mTORC1 and Myc in mLN GCB

To determine how Gα13-deficiency promotes GC expansion and proliferation in the mLN, we performed single-cell RNA sequencing (scRNA-seq) of GCBs from mLNs or immunized pLNs of WT or Gα13-deficient animals. LZ signatures were enriched in clusters 0, 6, 7 and 16; DZ signatures were enriched in clusters 1, 2, 5 and 14 (Fig. 3a and Extended Data Fig. 4a,b). Clusters 3, 4 and 15 were expanded in cells from both mLN samples with a further expansion in the Gα13-deficient mLNs (Fig. 3b). Gene set enrichment analysis (GSEA) revealed that these three clusters were enriched for signatures associated with positive selection in the GC related to Myc and mTORC1 (refs. 2–4) and cell cycle progression such as E2F target genes which in the GC is related to increased cyclin D3 (Ccnd3) levels[26,27] (Fig. 3c and Extended Data Fig. 4c–e). Clusters 3 and 4 show a marked enrichment for expression of ribosomal genes consistent with GSEA (Extended Data Fig. 4f). Gene sets related to Myc and mTORC1 were enriched in both mLN samples with a greater enrichment in Gα13-deficient mLNs (Fig. 3d and Extended Data Fig. 4c–e,g,h). Although the oxidative phosphorylation (OXPHOS), unfolded protein response and PI3K–AKT–MTOR signaling signatures were enriched in clusters 3, 4 and 15, they were not enriched in the Gα13-deficient mLN samples compared to WT mLNs (Fig. 3d and Extended Data Fig. 4h). Collectively, these data show that signatures associated with positive selection and control of cell cycle progression are enriched in the mLN GC with a further upregulation induced by the loss of Gα13.

Myc protein expression is upregulated in a fraction of LZ GCBs undergoing positive selection before entry into the DZ[2,3]. To validate our gene expression findings, we stained mLN or pLN GCBs from control or Gα13-deficient mixed chimeras intracellularly for Myc. We found that the fraction of LZ cells expressing Myc protein was increased in Gα13-deficient mLN GCBs (Fig. 3e,f). Notably, Myc protein was not increased in Gα13-deficient pLN GCBs consistent with the

lack of outgrowths at this site (Fig. 3g). Additionally, we found increased Myc expression in situ in Gα13-deficient mLN GCBs (Fig. 3h,i). We also found increased expression of cyclin D3 in Gα13-deficient mLN GCBs consistent with the differential expression of E2F target signatures (Extended Data Fig. 4i); however, unlike Myc, cyclin D3 was upregulated in Gα13-deficient GCBs in both mLNs and pLNs (Extended Data Fig. 4j). As signatures associated with mTORC1 signaling were increased in Gα13-deficient mLN GCBs, we measured phospho-ribosomal protein S6 (pRPS6), an important output of mTORC1, by flow cytometry and in situ. pRPS6 was increased in Gα13-deficient LZ GCBs in mLNs but not in pLNs (Fig. 3j–n). These data suggest that microenvironmental cues in the mLN may be important for supporting mTORC1 and Myc in Gα13-deficient GCBs.

## Gα13 signaling represses MYC translation

To better understand Gα13 signaling, we developed three GCB-DLBCL cell line models stably expressing doxycycline-inducible Cas9 (NUDUL1, OCI-Ly8 and Dogkit) in which we could induce Gα13 signaling. We transduced cell lines with the Gα13-coupled receptor Tbxa2r[28]. The Tbxa2r ligand, thromboxane-A2, is unstable and not present in cell culture medium[29]. U46619 is a stable analog of thromboxane-A2 and can induce Gα13 signaling when added to Tbxa2r-transduced cells[14,28] (Fig. 4a). We found that addition of U46619 led to loss of viable cells over time (Fig. 4b). Notably, when cells were transduced with a retrovirus expressing *GNA13* sgRNA and GFP, *GNA13* sgRNA+ (GFP+) cells outcompeted non-transduced cells in the presence of U46619 but did not show a competitive advantage in the absence of Gα13 stimulation (Fig. 4c and Extended Data Fig. 5a,b). Addition of U46619 led to a Gα13-dependent reduction in cells in S-G2 (Fig. 4d). Gα13 signaling suppressed MYC and E2F gene signatures (Fig. 4e).

Notably, U46619 treatment induced Gα13-dependent loss of MYC and cyclin D3 protein (Fig. 4f). Because mTORC1 signaling was increased in Gα13-deficient mLN GCBs, we evaluated whether Gα13 signaling could inhibit mTORC1. U46619 treatment resulted in sustained inhibition of the phosphorylation of mTORC1 targets P70 S6 kinase (pP70 S6K) and RPS6 (Fig. 4g). While Akt phosphorylation at T308 and S473 was also reduced by Gα13 signaling, its inhibition

**Fig. 4 | Gα13 signaling regulates multiple proliferative pathways. a**, GCB-DLBCL cell lines were engineered to express Cas9 and thromboxane receptor (Tbxa2r) and the synthetic thromboxane-A2 mimetic U46619 was added to cells to stimulate Gα13 signaling. **b**, Tbxa2r-expressing GCB-DLBCL cell lines were treated with the U46619 and the percent of live cells in culture was measured over time. Data are representative of two experiments for each cell line. **c**, Relative frequency of control or *GNA13* sgRNA-expressing NUDUL1 cells with or without U46619. Data are representative of four experiments. **d**, DNA content of control or *GNA13* sgRNA-expressing NUDUL1 cells treated with U46619 for 24 h. Data are representative of two experiments. **e**, GSEA of control or *GNA13*-deficient NUDUL1 cells treated for 24 h with U46619. **f**, MYC or cyclin D3 protein expression in control or GNA13-deficient GCB-DLBCL cells following U46619 treatment for 3 h or 6 h. Data are representative of two independent experiments for each cell line. **g**, Time course of phospho-P70S6K T389, phospho-RPS6 S235, phospho-Akt

S473 and T308 and MYC expression in U46619-treated NUDUL1 cells. Data are representative of three experiments. **h,i**, Experimental scheme (**h**) or CRISPR screen scores (CSSs) (**i**) for a genome-wide CRISPR/Cas9 screen for effectors of Gα13 signaling in GCB-DLBCL cell lines. Inducible Cas9 and Tbxa2r-expressing NUDUL1 or OCI-Ly8 cells were transduced with the genome-wide Brunello sgRNA library, following selection for transduced cells, Cas9 was induced with doxycycline for 7 days, cells were then cultured in the presence or absence of U46619 for 14 days and sgRNA representation was assessed by next-generation sequencing. **j,k**, Relative frequency of *ARHGEF1* (**j**) or *RIC8A* (**k**) sgRNA-expressing NUDUL1 cells with or without U46619. Data are representative of at least three independent experiments. **l**, Ratio of sgRNA-expressing Cas9+ GCBs (Cas9+sgRNA+ GCBs) to Cas9+ sgRNA+ FoBs in constitutive Cas9 BM chimeras targeting *Ric8* generated as shown in Extended Data Fig. 6c. Data are from one experiment $n = 5$. **$P = 0.008$ unpaired two-tailed Student's $t$-test.

was more short-lived compared to mTORC1 outputs, suggesting differential regulation of these pathways. Finally, to identify what outputs of Gα13 signaling were required to suppress cell expansion, we performed whole-genome CRISPR/Cas9 screens in Tbxa2r and

inducible Cas9-expressing NUDUL1 and OCI-Ly8 cell lines treated with U46619 (Fig. 4h). We identified only three genes as essential for U46619-mediated suppression of cell survival: *GNA13* itself, *ARHGEF1*, a Rho guanine exchange factor (GEF) that functions immediately

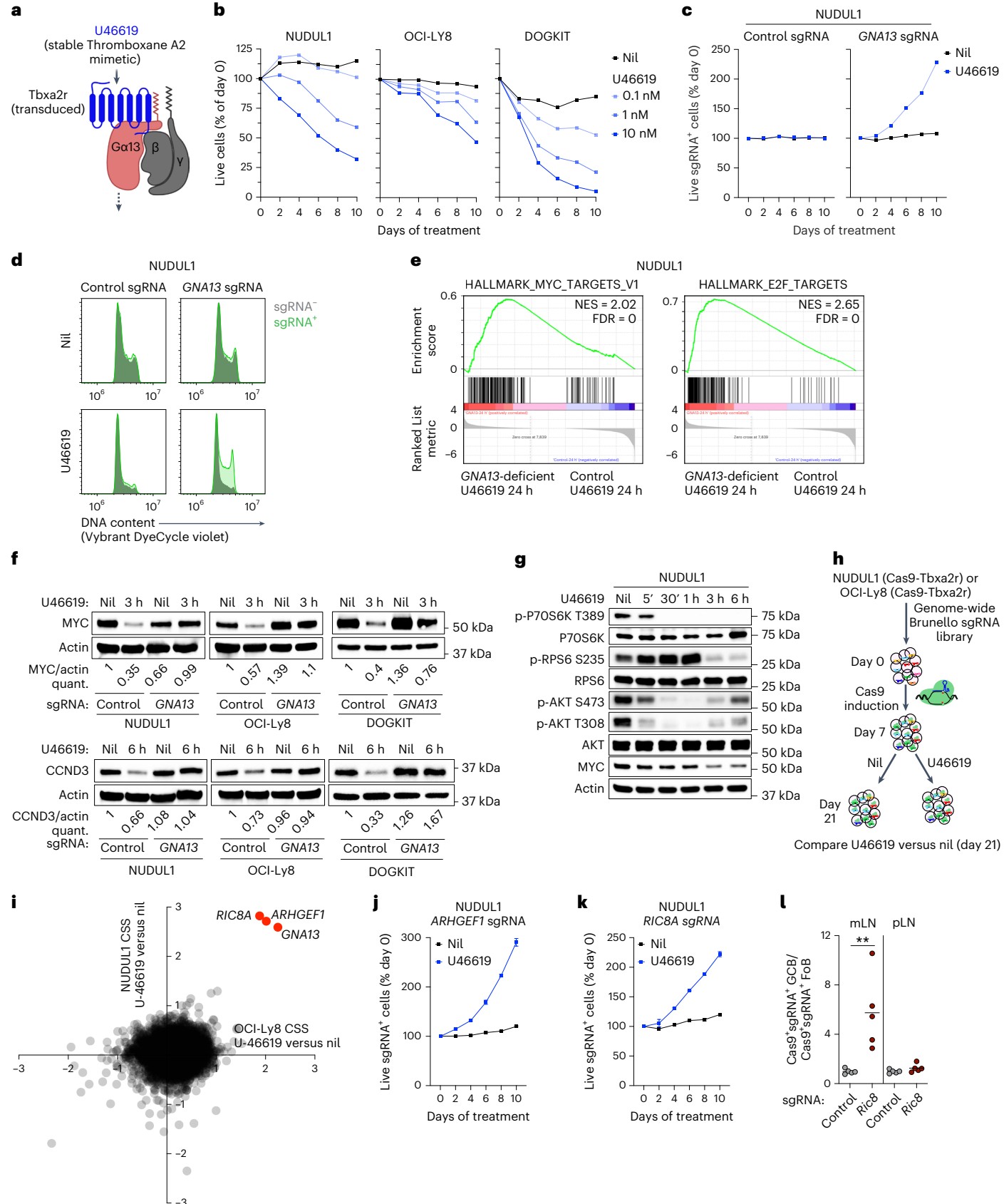

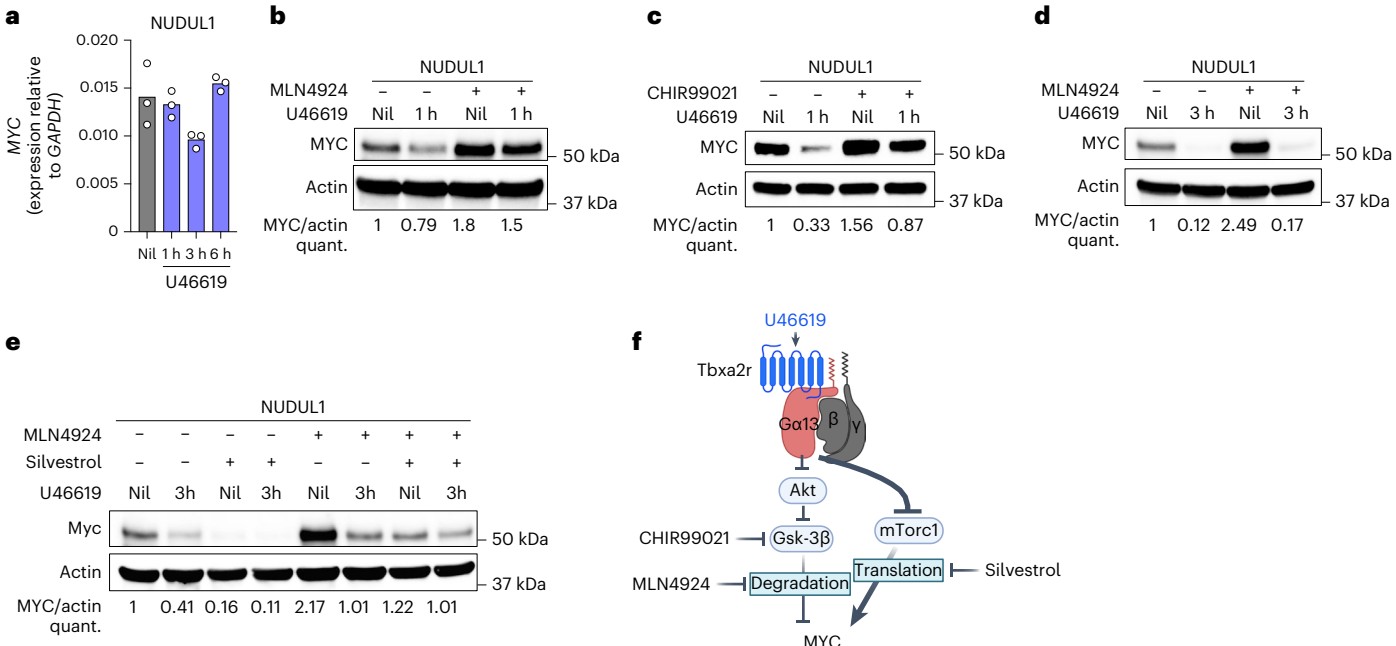

**Fig. 5 | Gα13 signaling suppresses MYC translation. a,** Quantitative PCR of *MYC* in NUDUL1 cells treated with U46619. Each point represents a technical replicate from one experiment representative of two. **b,c,** MYC protein expression in NUDUL1-Tbxa2r cells treated with U46619 and the neddylation inhibitor, MLN4924 (**b**) or the GSK-3b inhibitor, CHIR99021 (**c**) for 1 h. Data are representative of three and two experiments, respectively. **d,e,** MYC protein expression in NUDUL1 cells treated with U46619 in the presence or absence of MLN4924 (**d**) or the translation inhibitor, silvestrol (**e**) for 3 h. Data are representative of three and two experiments, respectively. **f,** Cartoon depicting Gα13-mediated regulation of MYC protein expression.

downstream of Gα13 and is critical for Gα13 signaling in GCBs[7,14] and *RIC8A*, a known GEF and chaperone for multiple Gα protein subunits[30] (Fig. 4i). *ARHGEF1* and *RIC8A* sgRNA-transduced cells outcompeted untransduced cells in the presence of U46619 (Fig. 4j,k). Additionally, Cas9-targeting of *Ric8*, the mouse homolog of *RIC8A*, in BM chimeras induced a selective outgrowth of GCB in the mLN confirming that Ric8 is preferentially involved in Gα13 signaling in vivo (Fig. 4l and Extended Data Fig. 5c). As we only identified molecules involved in proximal GPCR signaling and did not identify any single signaling pathway as critical for suppression of cell expansion, we conclude that Gα13 acts as a central regulator of multiple pathways essential for cell cycle progression and metabolism.

We next sought to understand how MYC is lost following Gα13 stimulation. *MYC* mRNA was only slightly reduced following U46619 suggesting that Gα13 primarily regulates MYC post-transcriptionally (Fig. 5a). To evaluate the contribution of proteasomal degradation to Gα13-mediated loss of MYC, we treated cells for 1 h with U46619 with the neddylation inhibitor MLN4924, which inhibits cullin-dependent proteasome-mediated degradation. We found that loss of MYC at 1 h following Gα13 stimulation was partially dependent on proteasomal degradation (Fig. 5b). MYC is targeted for proteasome-mediated degradation following phosphorylation by GSK-3B (ref. 31). Treatment of cells with the GSK-3B inhibitor CHIR99021 partially prevented U46619 induced MYC loss at 1 h (Fig. 5c); however, at 3 h following Gα13 stimulation, MYC was reduced even when the proteasome was inhibited (Fig. 5d). We then treated cells with U46619 and MLN4924 to block degradation with silvestrol, an inhibitor of translation[32]. We found that in the presence of silvestrol and MLN4924, the magnitude of Gα13-induced loss of MYC was reduced (Fig. 5e). Collectively, these data suggest that sustained Gα13-mediated loss of MYC protein is partially due to reduced translation (Fig. 5f).

*CCND3* mRNA was only slightly reduced following Gα13 stimulation (Extended Data Fig. 5d). Gα13-mediated loss of cyclin D3 was dependent primarily on proteasomal degradation (Extended Data Fig. 5e,f). Cyclin D3 is phosphorylated at T283 targeting it for

proteasomal degradation[33]. Gα13 stimulation increased p-T283 cyclin D3 that was accentuated in the presence of MLN4924 (Extended Data Fig. 5g). Whereas MYC loss was primarily mediated by reduced translation, these data suggest that Gα13 signaling reduces cyclin D3 primarily by promoting degradation.

## Dietary nutrients support Gα13-deficient GCBs in mLNs

Because MYC and mTORC1 gene signatures were increased in Gα13-deficient mLN GCBs and because Gα13 signaling promoted MYC loss in part by reducing its translation, we evaluated the contribution of mTORC1 signaling to expansion, Myc expression and proliferation in Gα13-deficient cells. In vitro, inhibition of mTORC1 by rapamycin led to a partial reduction in the competitive advantage of Gα13-deficient cells in the presence of U46619 (Fig. 6a). In vivo, we found that increased Myc and enhanced proliferation of Gα13-deficient mLN GCBs was partially dependent on mTORC1 (Fig. 6b–d). In immunized settings, T follicular helper cells are a critical driver of mTORC1 activation and Myc expression[2–4]. To assess whether T cell help in the mLN could drive differential expansion of Gα13-deficient GCBs, we depleted CD4[+] cells in Gα13-deficient mixed chimeras. We found that there was a 70% reduction in WT mLN GCBs following CD4[+] T cell depletion compared to a 50% reduction for Gα13-deficient cells, suggesting that expansion of Gα13-deficient GCBs in mLNs was not due to greater T cell help (Fig. 6e and Extended Data Fig. 6a).

As we considered mLN-specific factors that might support the local outgrowth of Gα13-deficient GCBs, we asked about the possible role of intestinal lymph-derived molecules. Specifically, we speculated that due to their reduced confinement to the GC and greater access to the follicular mantle and subcapsular lymphatics Gα13-deficient GCBs in the mLN may be supported by gut-derived factors delivered via lymph. Consistent with this hypothesis, we could identify Myc[+] GCBs near and within the subcapsular sinus of mLNs from Gα13-deficient animals (Fig. 6f). We have previously demonstrated that the pro-migratory S1P receptor, S1pr3, promotes access of Gα13-deficient GCBs to the outer follicle and lymph[34]. To assess

whether access to lymph-derived factors uniquely supports the survival of Gα13-deficient GCBs in mLNs, we targeted *S1pr3* in Aid-Cas9 or Aid-Cas9 *Gna13*^f/f BM chimeras. We found that S1pr3 expression supported the competitive advantage of Gα13-deficient GCB in the mLN and had no effect on WT GCBs or Gα13-deficient cells in other organs (Fig. 6g and Extended Data Fig. 6b). We then asked whether cues from gut microbiota could drive outgrowths of Gα13-deficient GCBs in mLNs. We first treated Gα13-deficient mixed chimeras with broad-spectrum antibiotics in drinking water. Antibiotic treatment led to increases of both WT and Gα13-deficient GCBs in mLNs; however, the competitive advantage of Gα13-deficient mLN GCBs was not affected by depletion of microbiota (Fig. 6h and Extended Data Fig. 6c). Given that there could be off-target effects from antibiotic treatment, we rederived *Cr2-cre Gna13*^f/f animals into germ-free conditions and assessed mLN GCB numbers and Myc expression. Germ-free Gα13-deficient mice showed expansion of mLN GCBs and increased Myc expression in LZ GCBs (Fig. 6i,j and Extended Data Fig. 6d). The increase in GCBs in germ-free animals was of similar magnitude to the increase seen in specific-pathogen-free Gα13-deficient animals (compare to Fig. 1d). Collectively, these data show that gut microbiota are not required for expansion of Gα13-deficient mLN GCBs.

Because mLNs are exposed to lymph derived from the small intestine, and mTORC1 is critically involved in nutrient sensing, we evaluated whether dietary nutrients could differentially support outgrowths of Gα13-deficient GCBs. We fasted Gα13-deficient mixed chimeras of food for 24 h and found that Gα13-deficient GCBs were lost to a greater extent than WT GCBs in the mLN (Fig. 6k and Extended Data Fig. 6e). We then sought to determine whether specific dietary nutrients could differentially support proliferation or expansion of Gα13-deficient GCBs in the mLN. Diet-derived long chain fatty acids (FAs) are packaged into chylomicrons and transported via lymph to mLNs. Because recent work has described a role for FA oxidation (FAO) in supporting OXPHOS in GCBs[35], and mTORC1 could sense energy derived from FAO, we initially evaluated the role of FAO in supporting Gα13-deficient GCBs. We targeted *Cpt2*, which is required for FAO in GCBs[35,36], in Aid-Cas9 or Aid-Cas9 *Gna13*^f/f BM chimeras. Gα13-deficient GCBs in mLNs were insensitive to the loss of *Cpt2* in contrast to WT GCBs (Fig. 6l). Additionally, FAO inhibition with etomoxir did not reduce the competitive advantage of Gα13-deficient cells in the presence of U46619 (Extended Data Fig. 6f). We then evaluated whether transport of FAs into lymph supported Gα13-deficient GCBs in mLNs. We placed Gα13-deficient mixed chimeras on a diet in which the fat source consisted of medium-chain triglycerides (MCTs), which are not packaged into chylomicrons thereby bypassing the mLN[37]. Outgrowths of Gα13-deficient GCBs were not reduced in chimeras on an MCT diet (Extended Data Fig. 6g). Moreover, a high-fat diet led to a reduction of outgrowths of Gα13-deficient GCBs in mLNs (Extended Data Fig. 6g). Collectively, these data show that while dietary fat and FAO plays a role in supporting WT mLN GCBs, they do not support

the outgrowth of Gα13-deficient GCBs. While in the LZ, GCBs have increased uptake of glucose and are thought to rely on glycolysis to meet energetic demands[38,39]. Therefore, we evaluated whether Gα13-deficient GCBs were more reliant on glucose than WT. First, in Aid-Cas9 chimeras, Gα13-deficient GCBs in mLNs seemed less dependent on expression of the glucose transporter *Slc2a1* (GLUT1) in vivo compared to WT (Fig. 6l). In vitro, we found that Gα13-deficient cells were more competitive than WT in the presence of the glycolysis inhibitor 2-deoxyglucose or in medium lacking glucose and pyruvate, suggesting that Gα13-deficient cells are not differentially reliant on glucose availability (Extended Data Fig. 6h,i). The sensing of amino acids is required for optimal activation mTORC1 (ref. 5). Most amino acids, except for glutamine and asparagine, are sensed via Rag GTPase-dependent pathways[40,41]. We evaluated Rag-dependent pathways by deleting *RRAGA* (RagA). In competition assays, deletion of *RRAGA* was similarly toxic in the presence or absence of Gα13 stimulation (Extended Data Fig. 6j). Additionally, deletion of *Rraga* in GCB in vivo resulted in a similar loss of both WT and Gα13-deficient mLN GCBs (Fig. 6l).

Glutamine is a potent activator of mTORC1 that acts in a Rag GTPase-independent manner[40,41]. Myc expression can also promote glutamine catabolism creating a positive feedback loop creating increased dependence on glutamine[42]. Recent work has suggested that restriction of the availability of glutamine may contribute to GC shutdown in some contexts[43]. We found that mTORC1 activation and MYC expression in GCB-DLBCL cell lines were highly dependent on the presence of glutamine in cell culture medium (Fig. 7a). MYC expression was decreased in the absence of glutamine thereby blunting the ability of U46619 to further reduce MYC (Fig. 7a and Extended Data Fig. 7a). The competitive advantage of Gα13-deficient cells in the presence of U46619 was also reduced when cells were cultured without glutamine (Fig. 7b). mTORC1 activation and MYC expression were dynamically regulated by the presence of glutamine in vitro. When cells were cultured in the absence of glutamine, pRPS6 and MYC were significantly reduced but could be restored by the addition of glutamine for 1 h and 3 h, respectively (Fig. 7c). In the presence of Gα13 stimulation, addition of glutamine to glutamine-starved cells was not able to restore Myc expression (Extended Data Fig. 7a). To determine whether dietary glutamine could support increased proliferation of Gα13-deficient mLN GCBs, we placed Gα13-deficient mixed chimeras on diet lacking glutamine as well as glutamic acid to prevent de novo generation of glutamine from glutamic acid and asparagine[44]. We found that glutamine derived from dietary sources supported increased pRPS6, increased Myc expression and increased proliferation of Gα13-deficient mLN GCBs (Fig. 7d and Extended Data Fig. 7b–d). We then sought to determine whether Gα13-deficient mLN GCBs were dependent on the expression of specific glutamine transport proteins[45]. *Slc38a1* encodes an important glutamine transporter that is highly and broadly expressed in GCBs (Extended Data Fig. 7e,f)[46]. We targeted *Slc38a1* in

**Fig. 6 | Dietary factors support Gα13-deficient mLN GCBs. a**, Frequency of *GNA13* sgRNA+ NUDUL1 cells treated with U46619 and rapamycin. Data are from one experiment representative of three. **P = 0.0089, ***P = 0.0005, ***P = 0.0003, unpaired two-tailed Student's *t*-test. **b–d**, Experimental scheme (**b**), percentages of Myc+ LZ (**c**) or BrdU+ (**d**) GCBs in mLNs of mixed chimeras that were treated with rapamycin. Data are from two experiments, n = 6 vehicle, n = 7 rapamycin. **P = 0.0017, *P = 0.0245, **P = 0.0012, paired two-tailed Student's *t*-test, ***P = 0.0007, *P = 0.0139, *P = 0.005, unpaired two-tailed Student's *t*-test. **e**, Ratio of CD45.2 GCBs to CD45.2 FoBs or frequency of GCBs in mLNs of mixed chimeras depleted of CD4+ cells. Data are from two experiments, n = 10. ***P = 0.0003, *P = 0.0459, unpaired two-tailed Student's *t*-test. **f**, Confocal microscopy of mLNs stained for LYVE1, IgD, GL7 and Myc. Scale bars, 100 μm (whole GC) or 25 μm (inset). Data are representative of three experiments, n = 3. **g**, Ratio of sgRNA+ GCBs to sgRNA+ FoBs in Aid-Cas9 or Aid-Cas9 *Gna13*^f/f BM chimeras targeting *S1pr3*. Data are from two experiments, n = 10. **P = 0.0035, unpaired two-tailed Student's *t*-test. **h**, Ratio of CD45.2 GCBs to CD45.2 FoBs

or frequency of GCBs in mLNs of mixed chimeras treated with antibiotics (ampicillin, vancomycin, neomycin, metronidazole; AVNM). Data are from two experiments, n = 18 control, n = 20 AVNM. ****P = 3.9 × 10⁻⁵, **P = 0.0065, unpaired two-tailed Student's *t*-test. **i,j**, Percentage of GCBs in mLNs (**i**) or Myc+ LZ GCBs (**j**) in germ-free animals. Data are from 12 experiments, n = 27 littermate, n = 14 Gα13-deficient. *P = 0.0296, unpaired two-tailed Student's *t*-test in **i**. *P = 0.0203, unpaired two-tailed Student's *t*-test in **j**. **k**, Ratio of CD45.2 GCBs to CD45.2 FoBs (middle) or frequency of GCB (right) in mLNs of mixed chimeras that were fasted of food. Data are from four experiments, n = 13 fed, n = 12 fasted. *P = 0.0263, *P = 0.0201, **P = 0.0028, unpaired two-tailed Student's *t*-test. **l**, Ratio of sgRNA+ GCBs to sgRNA+ FoBs in mLNs of Aid-Cas9 or Aid-Cas9 *Gna13*^f/f BM chimeras targeting *Cpt2, Slc2a1* or *Rraga*. Data are from three experiments, n = 15 control, n = 10 *Cpt2*, n = 9 *Slc2a1*, n = 4 *Rraga* (middle) and from four experiments, n = 20 control, n = 10 *Cpt2*, n = 13 *Slc2a1*, n = 4 *Rraga* (right). **P = 0.0055, **P = 0.0059, ***P = 0.0004, *P = 0.0425, ****P = 3.95 × 10⁻⁵, unpaired two-tailed Student's *t*-test.

Aid-Cas9 or Aid-Cas9 *Gna13*^f/f chimeras and found that Gα13-deficient GCBs in mLNs were dependent on *Slc38a1* in contrast to WT GCBs or Gα13-deficient GCBs in pLNs or PPs (Fig. 7e and Extended Data Fig. 7g).

We then evaluated whether excess dietary glutamine could differentially support expansion of Gα13-deficient GCB in mLNs. Addition of excess glutamine in drinking water for 3 weeks increased

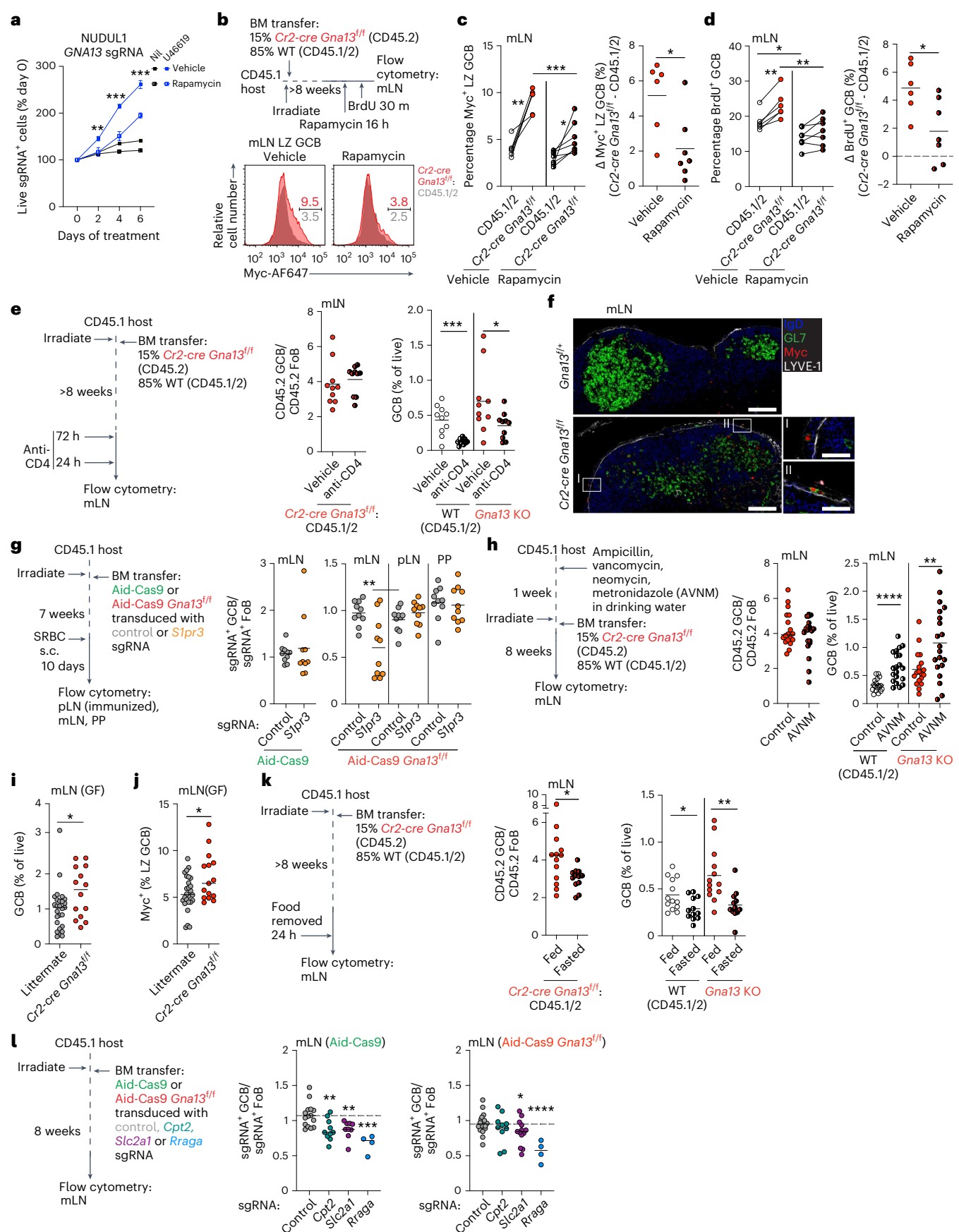

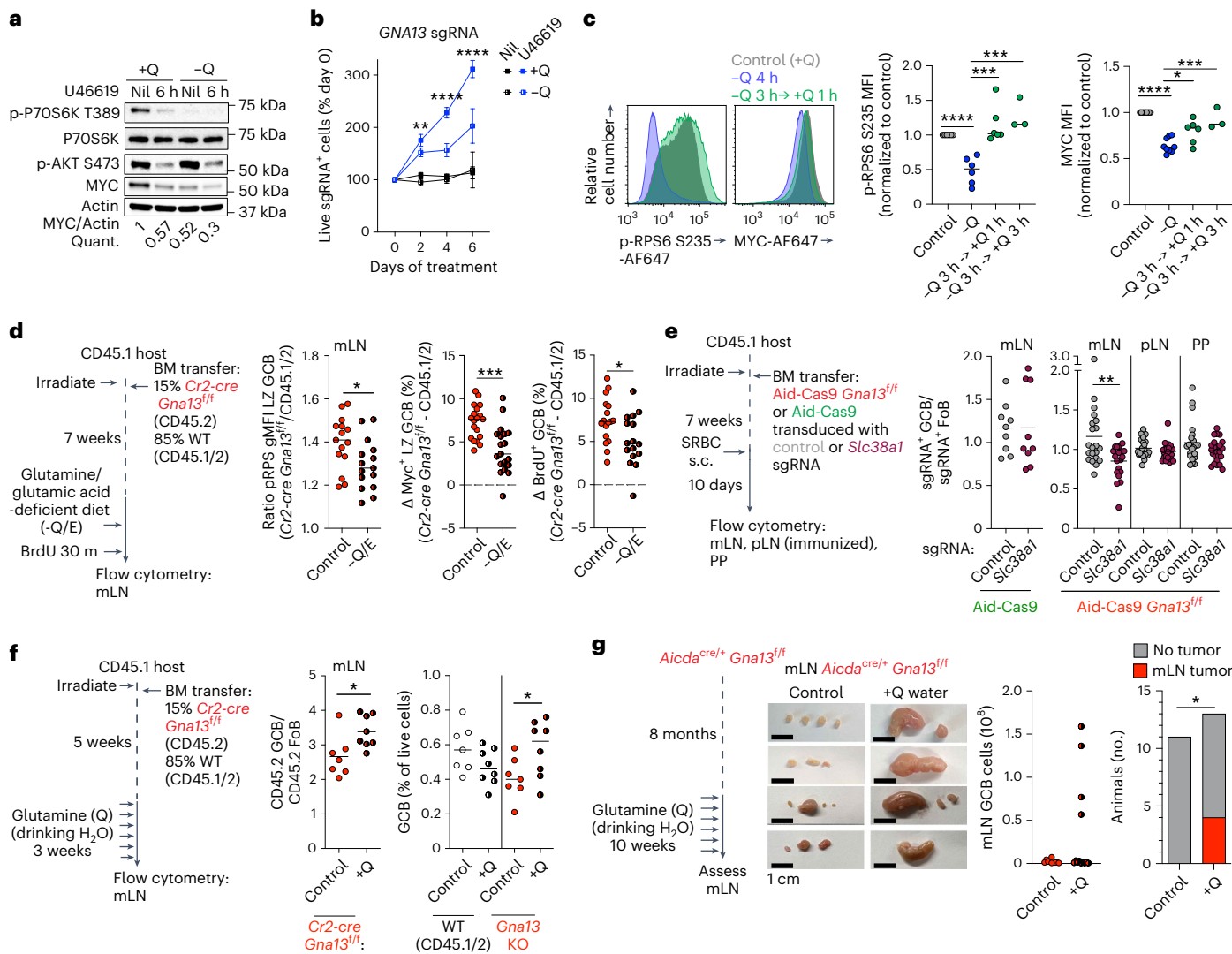

**Fig. 7 | Dietary glutamine differentially supports expansion of Gα13-deficient mLN GCBs. a**, Phospho-P70S6K T389, phospho-Akt S473 and MYC expression in NUDUL1 cells treated with U46619 with or without glutamine (Q). Representative of three experiments. **b**, Relative frequency of *GNA13* sgRNA⁺ NUDUL1 cells treated with U46619 with or without glutamine. Representative of three experiments. **P = 0.0023, ****P = 7.5 × 10⁻⁷ and 2.83 × 10⁻⁵, respectively, unpaired two-tailed Student's *t*-test. **c**, pRPS6 and MYC expression in NUDUL1 cells cultured without glutamine for 4–6 h or without glutamine for 3 h with glutamine add back for 1–3 h. Data are pooled from four experiments. ****P = 6.5 × 10⁻⁵, ***P = 0.0006 or 0.0008 and ****P = 3.4 × 10⁻¹⁰, *P = 0.0141 or ***P = 0.0007, unpaired two-tailed Student's *t*-test. **d**, Ratio or differences in percentages between Gα13-deficient and WT of pRPS6 gMFI or Myc⁺ in LZ GCB or BrdU⁺ GCBs in mLNs of mixed chimeras given glutamine and glutamic acid-

deficient diet (−Q/E diet). Data are pooled from seven experiments, n = 15, 15, 19, 19, 17 and 16. *P = 0.0225, ***P = 0.0003, *P = 0.0256 unpaired two-tailed Student's *t*-test. **e**, Ratio of sgRNA⁺ GCBs to sgRNA⁺ FoBs in Aid-Cas9 or Aid-Cas9 *Gna13*^f/f BM chimeras targeting *Slc38a1*. Data are pooled from two experiments, n = 9 and from four experiments, n = 21 control, n = 20 *Slc38a1*. **P = 0.0061, unpaired two-tailed Student's *t*-test. **f**, Ratio of CD45.2 GCBs to CD45.2 FoBs or frequency of GCBs in mLNs of mixed chimeras given glutamine for 3 weeks. Data are pooled from two experiments, n = 7 control, n = 8 glutamine. *P = 0.022 (middle), *P = 0.042 (right), unpaired two-tailed Student's *t*-test. **g**, Images of mLNs, GC B cell number or tumor incidence in Gα13-deficient animals given glutamine at 8 months of age for 10 weeks. Data are pooled from three experiment, n = 11 control, n = 13 glutamine. *P = 0.0219, one-sided chi-squared test.

---

both the competitive advantage and total number of Gα13-deficient GCBs in mLNs but not pLNs (Fig. 7f and Extended Data Fig. 7h,i). Finally, we assessed whether long-term glutamine supplementation could synergize with loss of Gα13 to promote tumor formation. We treated 8-month-old control (*Aicda*^cre/+*Gna13*^+/+) mice or mice lacking Gα13 in GCBs (*Aicda*^cre/+*Gna13*^f/f) with glutamine water for 10 weeks. Four of 13 Gα13-deficient mice treated with glutamine developed massive mesenteric lymphadenopathy with a greater than 25-fold expansion of GCBs (Fig. 7g). We did not observe any lymphadenopathy in the pLNs of Gα13-deficient animals placed on long-term glutamine. Additionally, Gα13-sufficient mice treated with glutamine did not develop enlarged mLNs or GC expansion (Extended Data Fig. 7j). These data

show that dietary glutamine promotes the proliferation and expansion of Gα13-deficient GCBs in the mLN via enhanced mTORC1 activation and MYC expression.

## Discussion

We demonstrate here that Gα13 is a tissue-specific tumor suppressor that controls GC proliferation in the mLN. Mechanistically, we found that Gα13 suppresses mTORC1 signaling and MYC protein expression, pathways usually associated with positive selection. Unexpectedly, differential expansion of Gα13-deficient GCBs was not dependent on T cell help. Moreover, we found no evidence that microbiota supported outgrowth of Gα13-deficient GCBs in the mLN, despite their anatomical

location. Instead, we determined that Gα13-deficient GCBs were differentially reliant on access to dietary nutrients delivered to the mLN via gut lymphatics—in particular glutamine. These insights help to explain the mLN tropism common to some aggressive lymphomas and offer clues to potential therapeutic vulnerabilities in these malignancies.

GCBs are normally tightly confined to the niche at the center of the follicle. This confinement is likely to limit access of GCBs to molecules that are delivered to lymph nodes via lymphatics as lymphatic conduits are relatively sparse inside follicles[47]. In this way the stringency of T cell-dependent GC-selective processes may be environmentally protected regardless of the lymphoid tissue in which the reaction is taking place. In the absence of Gα13, however, GCBs gain access to areas of the follicle more proximal to lymphatics and, therefore, may be exposed to nutrients or signaling molecules from which they are normally shielded. In this way, Gα13-deficient cells can bypass normal anatomic and signaling restrictions on GC-selective processes. We posit that access to these new areas of the mLN that are enriched with food-derived nutrients combined with the loss of Gα13-mediated inhibitory signaling on mTORC1 signaling and Myc expression may promote T cell-independent 'refueling' of Gα13-deficient GCBs, resulting in competitive expansion and clonal persistence in the GC state.

Aged mice lacking Gα13 formed primary tumors in mLNs but not in PPs and Gα13-deficient mixed chimeras showed larger outgrowths in mLNs than PPs. Although both PPs and mLNs are exposed to factors derived from gut microbiota, PP GCBs are greatly reduced in gnotobiotic animals whereas mLN GCBs are paradoxically increased, suggesting differential regulation of GCs at these two gut-associated sites[22]. We speculate that outgrowth and tumor formation are favored in the mLN because PPs lack afferent lymphatics and, in PPs, there is a larger anatomic separation between the GC and the subepithelial dome, the site at which gut luminal contents enter PPs, compared to the mLN GC and afferent lymphatics[48].

Although Gα13 signaling can inhibit PI3K/Akt signaling in GCBs[7,14,49], we demonstrate here that inhibition of PI3K/Akt is unlikely to be the primary factor driving the tumor suppressive activity of Gα13 in vivo. Instead, Gα13 signaling suppressed mTORC1 and MYC. MYC expression was suppressed primarily via reduced translation, likely via inhibition of mTORC1. Although in WT GCBs, MYC induction and mTORC1 signaling are thought to be initiated in parallel, with mTORC1 inhibition having only modest effects on MYC expression[4], we found that Myc expression in Gα13-deficient GCBs was strongly reduced in the presence of rapamycin, suggesting that increased mTORC1 activity is upstream of MYC expression in the absence of Gα13. Our data suggest that Gα13 signaling may selectively inhibit glutamine-induced activation of mTORC1, but further work is needed to define how this might occur.

Gα13 is frequently lost in BL and MYC-driven GCB-DLBCL[13]. BL most commonly presents with abdominal masses[17]. MLN involvement occurs in 30–50% of patients with non-Hodgkin lymphoma, of which DLBCL is the most common form[50]. Although no studies to date have examined the anatomic distribution of specific subtypes of DLBCL, it is likely that some Gα13-deficient tumors in humans develop initially from GCs present in the mLN and that dietary cues may drive premalignant expansions of Gα13-mutated GCBs. Our study suggests that targeting dietary nutrient-induced metabolic pathways can affect the development of disease in mice lacking Gα13 and that similar metabolic vulnerabilities exist in human lymphoma-derived cells. While future work is needed to determine whether dietary interventions can limit disease progression in animal models, our results highlight the possibility of dietary interventions as a rationally designed therapeutic strategy for patients with lymphoma.

## Online content

Any methods, additional references, Nature Portfolio reporting summaries, source data, extended data, supplementary information,

acknowledgements, peer review information; details of author contributions and competing interests; and statements of data and code availability are available at https://doi.org/10.1038/s41590-024-01910-0.

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

## Methods

### Animals

Adult B6-Ly5.1/Cr (B6.SJL-Ptprc$^a$Pepc$^b$/BoyCrCrl; stock no. 564) mice at least 6 weeks of age were obtained from Charles River Frederick Research Model Facility. *Cr2-cre* (B6.CgTg(Cr2-cre)3Cgn/J; stock no. 006368), *Aicda*$^{cre}$ (B6.129P2-*Aicda*$^{tm1(cre)Mnz}$/J; stock no. 007770), *Rosa26*$^{LSL-Cas9}$ (B6J.129(B6N)-*Gt(ROSA)26Sor*$^{tm1(CAG-cas9*,-EGFP)Fezh}$/J; stock no. 026175), *Rosa26*$^{Cas9}$ (B6(C)-*Gt(ROSA)26Sor*$^{em1.1(CAG-cas9*,-EGFP)Rsky}$/J; stock no. 028555) and Mb1-cre (B6.C(Cg)-*Cd79a*$^{tm1(cre)Reth}$/EhobJ; stock no. 020505) mice were from The Jackson Laboratory. *Gna13*$^{f/f}$ mice were from S. Coughlin[51] (University of California, San Francisco). *S1pr2-creERT2* BAC-transgenic and *Rosa26*$^{LSL-tdTomato}$ were previously described[21,52]. All mice were on a C57BL/6 background. Mice were housed in a specific-pathogen-free environment (except in gnotobiotic experiments) in ventilated microisolator cages with a 12-h light–dark cycle at 72 °F and 40–60% relative humidity. All mouse experiments received approval by the National Cancer Institute (NCI) Animal Care and Use Committee (ACUC) and were performed in accordance with NCI ACUC guidelines and under approved protocol LYMB-001. Male and female mice were used with an age range of 7–109 weeks. Littermates that were also sex-matched were used whenever possible to control for covariates. Mice were allocated to control and experimental groups randomly. No statistical methods were used to predetermine sample sizes, but our sample sizes are similar to those reported in previous publications[6,9,14,53]. Data collection and analysis were not performed blind to the conditions of the experiments. No data points were excluded from the analyses.

### Treatments

BM chimeras were made using B6-Ly5.1/Cr mice from Charles River as hosts. B6-Ly5.1/Cr hosts were lethally irradiated with a total of 900 rad in split doses. Hosts were later reconstituted by tail vein injection with at least $3 \times 10^6$ BM cells from the indicated donors. Mice were analyzed at least 8 weeks after reconstitution. Mice were injected with sheep red blood cells (SRBCs; Colorado Serum Company) s.c. to immunize pLN at the indicated time points. S1pr2-tdTomato mice were treated with 10 mg of tamoxifen (Sigma) dissolved in corn oil at 40 mg ml$^{-1}$ by oral gavage on days −2 and 0 and mice were analyzed at the indicated time points. Mice were treated i.p. with 62.5 µg of rapamycin (LC Laboratories) in ethanol at 10 mg ml$^{-1}$ that was freshly diluted in 250 µl PBS and analyzed 16 h later[4]. CD4$^+$ T cells were depleted with 250 µg anti-CD4 (GK1.5; Bio X Cell) i.p. given 72 h and 24 h before analysis. Gut microbiota were depleted in chimeras by treating mice with an antibiotic cocktail (1 g l$^{-1}$ ampicillin, 0.5 g l$^{-1}$ vancomycin, 1 g l$^{-1}$ neomycin and 1 g l$^{-1}$ metronidazole) in drinking water starting 1 week before irradiation and BM reconstitution. The antibiotic cocktail was made fresh and replaced twice weekly for the duration of the experiment. Unless otherwise noted, mice were fed a standard chow diet (NIH-31 5L7X; LabDiet). For fasting experiments, animals were transferred to fresh cages and food was removed for 24 h before analysis. For experiments modulating dietary fat intake, BM chimeras were placed on control diet (D12450J; Research Diets), high-fat diet (D12492; Research Diets) or MCT diet (D15052702; Research Diets) for 3 weeks before analysis. For experiments with diets lacking glutamine and glutamic acid, BM chimeras were placed on control L-amino acid rodent diet (A10021Bi; Research Diets) or L-amino acid rodent diet without added glutamine and glutamate (A20081701i; Research Diets) for at least 2 days before analysis. For experiments with glutamine supplementation, mice were treated with L-glutamine (28 g l$^{-1}$; Sigma) in drinking water for 3–10 weeks before analysis. The L-glutamine water was made fresh and replaced twice weekly for the duration of the experiment.

For gnotobiotic experiments, pregnant donor females that were *Cr2-cre* or *Gna13*$^{f/f}$ were injected s.c. with progesterone (5 mg ml$^{-1}$) on days e17.5 and e18.5 and the rederivation procedure was performed on gestational day e19.5 of the donor females. The donor female was killed by cervical dislocation and the uterus was submerged in a tube containing a warm Virkon solution and introduced into an isolator housing. The uterus was opened and the pups were fostered by germ-free Swiss foster mothers. Weaned animals were tested and confirmed to be free of any viruses, bacteria, fungi or parasites. Germ-free mice were bred and maintained in a sterile environment completely devoid of microorganisms in the Gnotobiotic Facility, Frederick National Laboratory for Cancer Research. NCI-Frederick is accredited by The Association for Assessment and Accreditation of Laboratory Animal Care International and follows the Public Health Service Policy for the Care and Use of Laboratory Animals.

### BM transduction

For transduction of BM, donor mice were injected intravenously with 3 mg 5-fluorouracil (Sigma). BM was collected after 4 days and cultured in DMEM containing 15% ($v/v$) FBS, antibiotics (penicillin (50 IU ml$^{-1}$) and streptomycin (50 mg ml$^{-1}$); Cellgro) and 10 mM HEPES, pH 7.2 (Cellgro), supplemented with interleukin (IL)-3, IL-6 and stem cell factor (at concentrations of 20, 50 or 100 ng ml$^{-1}$, respectively; Peprotech). Cells were 'spin-infected' twice with MRIA retrovirus[54] expressing non-targeting sgRNA (5′-AGCAGCGTCTTCGAGAGTG-3′), *Gna13* sgRNA (5′-TGCTATCAGAGCCTTATGGG-3′), *Pten* sgRNA (5′-CCTCCAATTCAGGACCCACG-3′ and 5′-TGTGCATATTTAT TGCATCG-3′), *Ric8* sgRNA (5′-TCTGCGGTCGTTCAACCGGG-3′ and 5′-ACAGGCATTTGAGAGACTCG-3′), *S1pr3* sgRNA (5′-GGGAA CATTACGATTACGTG-3′ and 5′-AATCACTACGGTCCGCAGAA-3′), *Cpt2* sgRNA (5′-TCACTGGTCAAATAAGCCAG-3′ and 5′-TCGGGAA GTCATCTAAGCAG-3′), *Slc2a1* sgRNA (5′-CCTGCTCATCAATCGTAACG-3′ and 5′-TCAGCATGGAGTTCCGCCTG-3′), *Rraga* sgRNA (5′-GGTTC CCCAAGAATCGGACG-3′ and 5′-GATCAGCTGATAGACGATGC-3′) or *Slc38a1* sgRNA (5′-ATACTTTGGTGTGCACGCGT-3′ and 5′-TGCAT GGTGTATGAGAAGCT-3′) that also expressed the fluorescent reporter ametrine or retrovirus in which myr-Akt or human CD4 (control) was downstream of a loxP–stop–loxP and also expressed Thy1.1 at days 1 and 2 and transferred into irradiated recipients on day 3.

### Flow cytometry and cell sorting

The pLN, mLN and PP cell suspensions were generated by mashing the organs through 70-mm cell strainers in RPMI containing 2% ($v/v$) FBS, antibiotics (penicillin (50 IU ml$^{-1}$) and streptomycin (50 mg ml$^{-1}$); Cellgro) and 10 mM HEPES, pH 7.2 (Cellgro). Cells were stained as indicated with the following antibodies and dyes: Fixable Viability Dye eFluor 780 (eBioscience; 1:1,000 dilution), BV786 or BUV395-conjugated anti-B220 (RA3-6B2; BD; 1:400 dilution), BUV395-conjugated anti-CD4 (RM4-5; BD; 1:400 dilution), Pacific blue or Alexa Fluor 647-conjugated GL7 (GL7; BioLegend; 1:400 dilution), BV650 conjugated anti-IgD (11-26c.2a; BioLegend; 1:400 dilution), PerCP-Cy5.5 or PE-Cy7-conjugated anti-CD38 (90; BioLegend; 1:400 dilution), PE-Cy7, PE or BV421-conjugated anti-Fas (Jo2; BD; 1:400 dilution), FITC, PerCP-Cy5.5 or Alexa Flour 700-conjugated anti-CD45.2 (104; BioLegend; 1:400 dilution), PE-Cy7 or PerCP-Cy5.5-conjugated anti-CD45.1 (A20; BioLegend; 1:400 dilution), BV786-conjugated anti-CD86 (GL-1; BioLegend; 1:200 dilution), PE-conjugated, APC-conjugated anti-CXCR4 (2B11; eBioscience; 1:200 dilution), Alexa Fluor 647-conjugated anti-c-Myc (Y69; Abcam; 1:400 dilution), PE-conjugated anti-cyclin D3 (DCS22; BioLegend; 1:400 dilution), pRPS6 Ser240/244 (D68F8; Cell Signaling; 1:400 dilution) pAKT S473 (D9E; Cell Signaling; 1:400 dilution) and/or AF647-conjugated anti-rabbit IgG (Invitrogen; 1:1,000 dilution). For BrdU incorporation experiments, animals were given 2.5 mg of BrdU in a single i.p. injection and killed 30 min later. Staining was performed using the FITC or APC BrdU Flow kit (BD Pharmingen) according to the manufacturer's instructions. In some BrdU experiments, FITC conjugated anti-BrdU (BU20A; eBioscience; 1:100 dilution) and DNase (D4513-1VL; Sigma) were used in place of reagents from BD. To stain for intracellular antigens (Myc, cyclin D3, pRPS6 and pAkt), cells were first stained

for surface markers and then fixed and permeabilized using the BD cytofix/cytoperm kit per manufacturer's instructions. For pRPS6 and pAkt, staining was precisely timed so that fixation/permeabilization occurred 30 min after killing to minimize time-related changes introduced during sample processing[4]. Flow cytometry was performed on a Cytoflex LX (Beckman Coulter). Flow cytometry data were collected with CytExpert v.2.3 (Beckman Coulter). Cells were sorted on a Sony MA900 sorter.

## RNA isolation, heavy chain sequencing
CD45.2[+] GCBs from *Gna13*[f/+] or *Cr2-cre Gna13*[f/f] mixed chimeras were sorted directly into RLT buffer (QIAGEN) and RNA was extracted according to the manufacturer's protocol. For assessment of heavy chain V gene usage, RNA from 8,000–42,000 sorted GCBs was sent on dry ice to iRepertoire. Complementary DNA synthesis, PCR amplification of heavy chain repertoire and analysis on an Illumina MiSeq Nano were performed with proprietary reagents by iRepertoire. Then, 184,000–1,664,000 reads were obtained for each sample.

## Single-cell RNA sequencing
GCBs from mLNs or pLNs were sorted from three *Gna13*[f/+] (WT) or three *Cr2-cre Gna13*[f/f] (KO) mice that were weeks 12 weeks old and immunized s.c. with SRBCs 10 days before killing. Approximately $1.5 \times 10^4$ cells for each condition were loaded on a 10x Chromium controller (10x Genomics) and libraries were constructed using the 5′ V2 reagents according to the manufacturer's instructions. WT pLN, WT mLN, KO pLN and KO mLN GCBs were captured separately in the same experiment. Libraries were sequenced on an Illumina NextSeq 2000 run using $26 \times 10 \times 10 \times 90$-bp sequencing. Sequencing files were processed and aligned to mm10 and count matrices were generated using Cell Ranger (v.6.0.0). Further analyses were performed in R using the Seurat package (v.4)[55]. GSEA was performed using the FGSEA workflow[56].

## Immunohistochemistry, immunofluorescence and histocytometry
The mLNs or mLN tumors were fixed in 4% paraformaldehyde (Electron Microscopy Sciences) and 10% sucrose in PBS for 1 h at 4 °C then moved to 30% sucrose in PBS overnight. Tissues were flash-frozen in OCT compound (Sakura) the following day. Then, 7-μM or 20-μM sections were cut on a Leica CM1950 cryostat and were adhered to Super Frost Plus slides (Fisher Scientific). For immunohistochemistry of mLN tumors, primary antibodies used for staining cryosections were biotinylated anti-GL7 (GL7; BioLegend; 1:400 dilution), biotinylated anti-B220 (RA3-6B2; BioLegend; 1:200 dilution), biotinylated anti-CD35 (8C12; BD Biosciences; 1:200 dilution) and unlabeled polyclonal goat anti-mouse IgD (Cedarlane; 1:1,000 dilution). Secondary antibodies or streptavidin fused to alkaline phosphatase or peroxidase were from Jackson Immunoresearch Laboratories. Images were captured using a Zeiss Axioscan Z1 slide scanner with Zen v.3.8 (Zeiss). For immunofluorescence, before staining, sections were fixed in ice-cold acetone for 10 min and then air dried for 1 h. Sections were blocked for 1 h in PBS containing 0.3% Triton X-100, 1% BSA, 2% normal mouse serum, 2% normal goat serum, 2% normal rat serum and 2% anti-CD16/32 (Bio X Cell). Sections were stained with SparkRed 718-conjugated anti-B220 (RA3-6B2; BioLegend; 1:200 dilution), AF488-conjugated anti-IgD (11–26c.2a; BioLegend; 1:100 dilution), AF647-conjugated GL7 (GL7; BioLegend; 1:100 dilution), biotin-conjugated anti-CD35 (8C12; BD; 1:200 dilution), biotin-conjugated anti-LYVE1 (ALY7, eBioscience; 1:200 dilution), rabbit anti-Myc (D3N8F; Cell Signaling; 1:400 dilution) and/or rabbit anti-phospho-ribosomal protein S6 (pRPS6) (Ser240/244) (D68F8; Cell Signaling; 1:800 dilution) overnight at 4 °C in a humidified chamber. Secondary antibodies were BV421-conjugated streptavidin (BD) and AF555-conjugated anti-rabbit IgG (Invitrogen; 1:200 dilution) and were incubated with slides for 3 h at 27 °C. Cell nuclei were stained with JOPRO-1 (Invitrogen; 1:10,000 dilution). Stained slides

were mounted with Fluoromount G (Southern Biotech) and sealed with a glass coverslip. Tile scans of mLNs were acquired using a Stellaris 5 confocal microscope (Leica) with LASX v.4.5 (Leica) with a ×63 oil immersion objective NA 1.4 at a voxel density of $2,056 \times 2,056$. For histocytometric analysis of Myc positivity and pRPS6 intensity in GCBs we stained mLN sections with a panel consisting of the following fluorophores: BV421, AF488, JOPRO-1, AF555, AF647 and SR718. Fluorophore emission was collected on separate detectors with sequential laser excitation. Surface rendering was performed as previously described using Imaris v.10 (refs. 57,58). Channel statistics for all surfaces were exported into Excel (Microsoft) and converted to a CSV file for direct visualization in FlowJo v.10. Single cells were identified on the basis of B220[+], IgD[+] and GL7[+] to identify FoB cells (B220[+]IgD[+]) and GCBs (B220[+]IgD[−]BCL6[+]). For quantification of Myc[+] in GCBs, GCBs were defined as Myc positive if their fluorescence was greater than three s.d. greater than the median fluorescence of FoB cells in each image. For quantification of pRPS6 in GCBs, gMFI of pRPS6 was normalized to the gMFI of FoB cells in each image.

## Cell culture
NUDUL1 was from ATCC CRL-2969, OCI-Ly8 was from NCI and Dogkit was from DMSZ. GCB-DLBCL cell lines were grown at 37 °C with 5% $CO_2$ in Advanced RPMI (Invitrogen) supplemented with 4% fetal bovine serum (Tet tested, Atlanta Biologics), 1% pen/strep (Mediatech), 10 mM HEPES, pH 7.2–7.6 (Corning) and 2 mM L-glutamine (Mediatech). The 293FT (Thermo) and PLAT-E (a gift from S. Schwab) cells were grown in DMEM (Invitrogen) supplemented with 10% fetal bovine serum, 1% pen/strep and 1% L-glutamine. Cell lines were regularly tested for *Mycoplasma* using the MycoAlert Mycoplasma Detection kit (Lonza).

## CRISPR screens
GCB-DLBCL cell lines were transduced with pCW-Cas9-BLAST (Addgene 83481), selected with blasticidin and dilution cloned. Single-cell clones of doxycycline-inducible Cas9-engineered GCB-DLBCL cells lines were first selected for exceptional exonuclease activity by testing knock-out efficiency with sgRNA directed against a surface marker. After selecting clones with strong Cas9 activity, cells were transduced with MSCV-Tbxa2r-IRES-Thy1.1 and selected for Thy1.1 expression by cell sorting. CRISPR screens were performed as previously described using the Brunello sgRNA library (Addgene, 73178)[59,60]. Following transduction with the Brunello library, cells were selected with puromycin for 4 days followed by doxycycline to induce Cas9 for 7 days. Pools of cells were then grown in the presence or absence of U46619 (Cayman) for an additional 14 days. DNA was extracted from cell pellets and libraries were generated from sgRNA sequences that were amplified from genomic DNA. sgRNA sequences were enumerated by next-generation sequencing using NextSeq 1000/2000 Control Software (v.1.2.036376) (Illumina) and demultiplexed using DRAGEN (v.3.7.4) (Illumina). CRISPR screen reads were extracted from fastq files and normalized using custom perl scripts and Bowtie2 (v.2.2.9). Detailed methods and PCR primer sequences have been previously described[61]. CRISPR screen scores comparing day 21 U46619 to day 21 nil are displayed in Fig. 4i for non-essential genes with a score greater than −1 comparing day 21 nil to day 0 samples.

## Single-gene KO experiments in cell lines
For single-gene experiments, gRNA sequences were cloned into pLKO.1-Puro-GFP as described previously[59]. GCB-DLBCL cell lines were infected with a concentrated virus and 3 days after transduction, doxycycline was added to induce Cas9 expression and 1 week later cells were used in additional experiments. The following sgRNAs were used in this study: sgControl: 5′-TAAAGCAGAAGAATATACAG-3′; sg*GNA13*: 5′-AGAGATCAGAAAGGAAACGT-3′, sg*ARHGEF1*: 5′-TGGA GGACTTCCGTTCCAAG-3′, sg*RIC8A*: 5′-CAGTACAACATCCATGTCTG-3′ and sg*RRAGA*: 5′-GCTGAACGTTGGGAATCAGC-3′.

## Competitive survival assays

For competitive survival assays, sgRNA-transduced (GFP⁺) cells were mixed with untransduced control cells in a 1:4 ratio and measured by FACS over a 6–10-day period every 2 days. Cells were treated with U46619 (10 nM; Cayman), rapamycin (50 nM; Cayman), etomoxir (10 μM; Cayman), 2-deoxyglucose (1 mM; Cayman) or in medium lacking glutamine (Advanced RPMI with pen/strep, HEPES and 4% FCS) or lacking glucose and pyruvate (10-043-CV; Mediatech).

## Cell cycle assessment

Cell cycle analyses of sgRNA-transduced cells were performed using Vybrant Dye Cycle Violet (Thermo) and assessed by FACS.

## Immunoblotting

Cells with treated in the with U46619 (10 nM; Cayman), silvestrol (25 nM; Biovision), MLN4924 (1 μM; Cayman) and/or CHIR99021 (1 μM; Sigma) for the indicated time points and then washed in ice-cold PBS and lysed in ice-cold RIPA buffer (Cell Signaling) and complete protease inhibitor cocktail (Roche) for 10–30 min on ice. Lysates were cleared by centrifugation at 17,000 $g$ at 4 °C for 15 min and post-nuclear supernatant was collected and boiled for 10 min in Laemmli sample buffer (Bio-Rad) supplemented with β-mercaptoethanol (Bio-Rad). Samples were run on Any kD Mini Protean TGX Precast Gels (Bio-Rad), transferred to a PVDF membrane, blocked in 5% milk in TBST (pH 7.4) and incubated with the following primary antibodies in 5% milk overnight: MYC (D3N8F; Cell Signaling; 1:2,000 dilution), CCND3 (DCS22; Cell Signaling; 1:2,000 dilution), actin (13E5; Cell Signaling; 1:4,000 dilution), p-P70S6K T389 (108D2; Cell Signaling; 1:2,000 dilution), P70S6K (polyclonal rabbit antibody 9202; Cell Signaling; 1:2,000 dilution), pRPS6 S235/6 (D57.2.2E; Cell Signaling; 1:4,000 dilution), RPS6 (5G10; Cell Signaling; 1:4,000 dilution), pAKT S473 (D9E; Cell Signaling; 1:4,000 dilution), pAKT T308 (D25E6; Cell Signaling; 1:4,000 dilution) and p-CCND3 T283 (E1V6W; Cell Signaling; 1:2,000 dilution). Membranes were washed 3× in TBST (pH 7.4), probed with anti-rabbit-HRP or anti-mouse-HRP antibodies (CST) in 5% milk, washed again 3× in TBST (pH 7.4) and imaged on a ChemiDoc Imaging System (Bio-Rad) with Image Lab Touch Software v.3.01 (Bio-Rad). Images were analyzed with Image Lab v.6.1 (Bio-Rad).

## Quantitative PCR

Real-time PCR was performed with SYBR Green PCR Mix (Roche) and an ABI prism 7500 sequence detection system (Applied Biosystems). The following primers were used: *MYC* forward 5′-CCTTCTCT CCGTCCTCGGAT-3′, *MYC* reverse 5′-TTCTTGTTCCTCCTCAGAGTCG-3′, *CCND3* forward 5′-ACTGGCACTGAAGTGGACTG-3′, *CCND3* reverse 5′-GGGCTACAGGTGTATGGCTG-3′, *GAPDH* forward 5′-CGGAGTCA ACGGATTTGGTC-3′ and *GAPDH* reverse 5′-TCGCCCCACTTGA TTTTGGA-3′.

## RNA sequencing of cell lines

NUDUL1 cells expressing control or *GNA13*-targeting sgRNA were treated for 24 h with U46619. RNA was extracted using the RNeasy Mini kit (QIAGEN) and RNA libraries were prepared using the TruSeq Stranded mRNA Library Prep (Illumina) according to the manufacturer's protocol. Sequencing of libraries was conducted on a NovaSeq SP with a read length of 2 × 151 bp. Alignment to the human genome (hg19) was conducted using STAR aligner. Normalized reads were log₂-transformed to calculate digital gene expression values. GSEA was performed with GSEA software (v.4.0.3)

## Statistical analysis

Prism v.10 (GraphPad) and Microsoft Excel v.16.80 were used for statistical analysis. Data were analyzed with a two-sample unpaired (or paired, where indicated) Student's $t$-test. $P$ values were considered significant at ≤0.05. Data distribution was assumed to be normal, but this was not formally tested.

## Reporting summary

Further information on research design is available in the Nature Portfolio Reporting Summary linked to this article.

## Data availability

Cell line and scRNA-seq datasets have been deposited to the Gene Expression Omnibus and are available under accession code GSE253435. The heavy chain BCR repertoire sequencing and whole-genome CRISPR/Cas9 screens of GCB-DLBCL cell lines have been uploaded to figshare at https://doi.org/10.6084/m9.figshare.25006655 (ref. 62). Source data are provided with this paper.

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

## Acknowledgements

We thank J. Cyster for providing resources and guidance in the initial stages of this project and for comments on the manuscript, L. Staudt, A. Reboldi, E.V. Dang, P. Chakraborty and A. Bolomsky for providing comments on the manuscript and Z. Coulibaly for help with data analysis. Brightfield imaging of tumor samples was performed in the Laboratory of Genitourinary Cancer Pathogenesis Microscopy Core Facility with assistance from R. Lake. Model figures were created with BioRender.com. This research was supported by the Intramural Research Program of the Center for Cancer Research, NCI, National Institutes of Health (J.R.M.). Gnotobiotic work was funded in part with federal funds from the NCI, National Institutes of Health, under contract no. HHSN261201500003I (S.D.).

## Author contributions

H.T.N. designed and performed experiments, analyzed and interpreted data, and wrote the manuscript. M.L. performed mouse genotyping, generated critical reagents and cared for the mouse colony. K.H. maintained the gnotobiotic mouse colony. T.C.T. analyzed scRNA-seq datasets. H.L. generated critical reagents and performed experiments. R.V., M.Y., S.D., E.G.C. and A.R.A. performed experiments and analyzed data. Y.Y. performed experiments. J.K. analyzed data. S.R. and D.W.H. analyzed data. M.K. performed experiments and analyzed data. J.D.P. and R.M.Y. analyzed data, provided critical input and edited the manuscript. S.P. analyzed and interpreted data. S.D. managed the gnotobiotic facility. J.R.M. designed and performed experiments, analyzed and interpreted data, and wrote the manuscript.

## Competing interests

The authors declare no competing interests.

## Additional information

**Extended data** is available for this paper at https://doi.org/10.1038/s41590-024-01910-0.

**Correspondence and requests for materials** should be addressed to Jagan R. Muppidi.

mLN tumors from *Cr2-cre Gna13^{f/f}* (14.5-18 months)

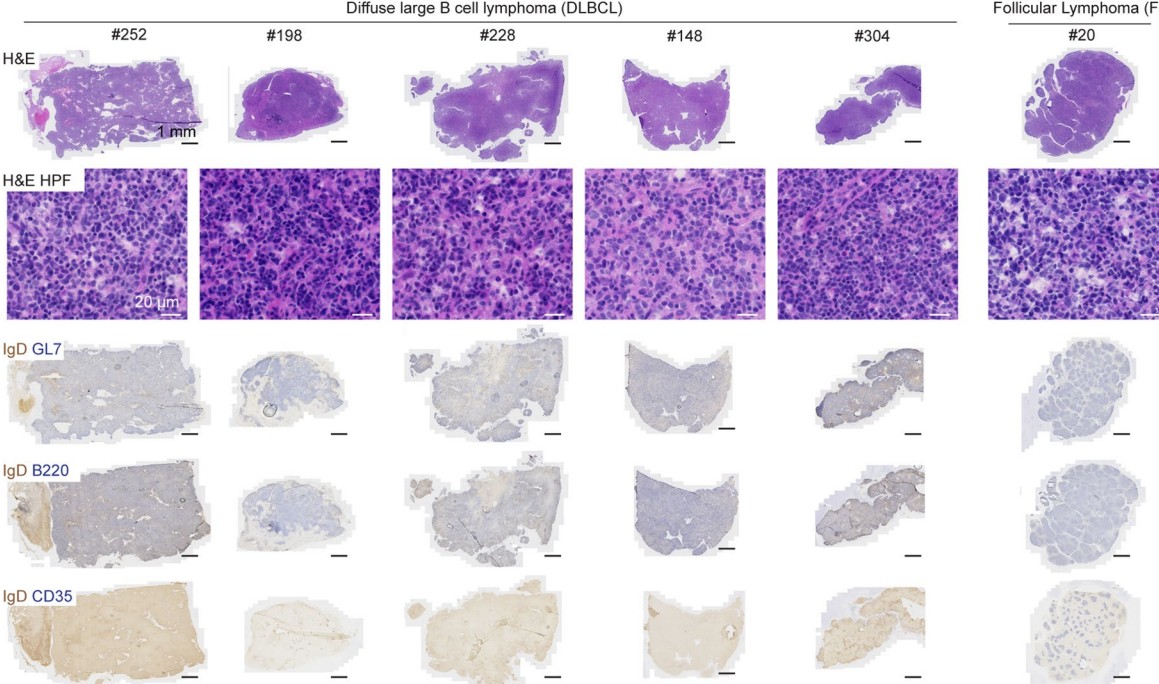

**Extended Data Fig. 1 | Gα13 suppresses B cell lymphoma development in the mLN.** Histological analysis of mLN tumors from *Cr2-cre Gna13^{f/f}* animals aged 14–18 months. Sections were stained with hematoxylin and eosin (H&E) or for IgD and GL7, B220 or CD35. Scale bars: 1 mM in low power H&E and immunohistochemistry images and 20 μm in high power H&E (HPF). Data are from 6 tumors representative of 11 tumors.

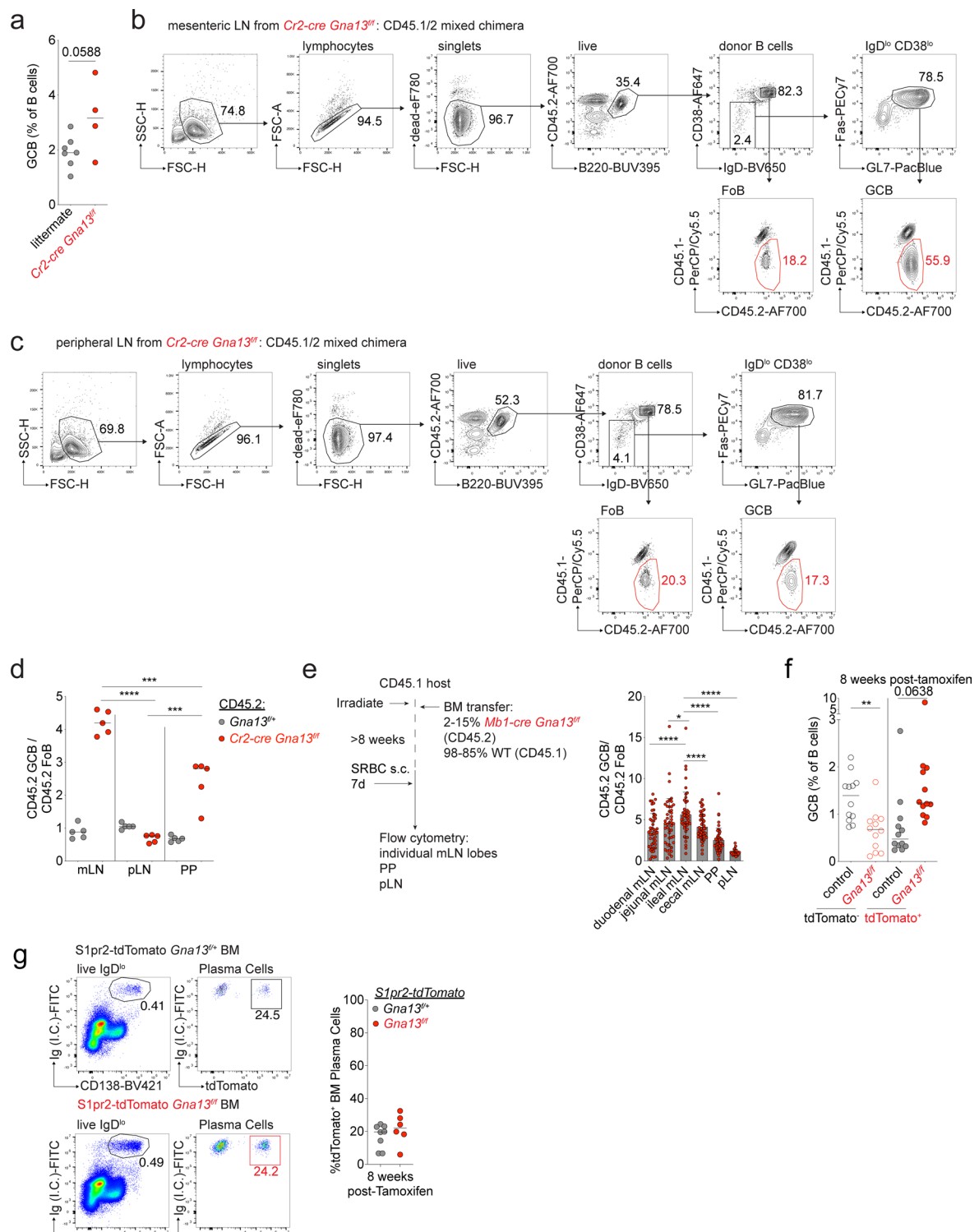

**Extended Data Fig. 2 | Gα13 suppresses cell intrinsic GC B cell expansion in the mLN. (a)** GCB as a percentage of B cells in mLN of 8-week-old bred *Cr2-cre Gna13^{f/f}* or littermate control (*Gna13^{f/+}*, *Gna13^{f/f}* or *Cr2-cre Gna13^{f/+}*) animals. **(b, c)** Representative flow cytometric analysis of FoB and GCB in mLN (a) and pLN (b) of Gα13 mixed chimeras generated as in Fig. 1d. **(d)** Ratio of frequency of CD45.2 GCB to CD45.2 FoB in mLN, pLN and Peyer's patches (PP) of control or Gα13-deficient mixed chimeras generated as in Fig. 1d. One experiment representative of four, n = 5. ***P = 0.0009, ****P < 0.0001, ***P = 0.0005 unpaired two–tailed Student's *t*-test. **(e)** Ratio of frequency of CD45.2 GCB to CD45.2 FoB

in individual lobes of the mLN chain draining the indicated portion of the small intestine, PP or SRBC-immunized pLN. 7 experiments, n = 47 for mLN and PP n = 24 for pLN. ****P < 0.0001, *P = 0.0495 paired two-tailed Student's *t*-test. **(f)** tdTomato^- or tdTomato^+ mLN GC B cells as a percentage of B cells in *S1pr2-tdTomato* control or *Gna13^{f/f}* mice at 8 weeks following tamoxifen administration. **P = 0.0015 unpaired two-tailed Student's *t*-test. **(g)** Percentage of tdTomato^+ BM plasma cells in *S1pr2-tdTomato* control or *Gna13^{f/f}* mice at 8 weeks following tamoxifen administration. 2 experiments, n = 8 and 6.

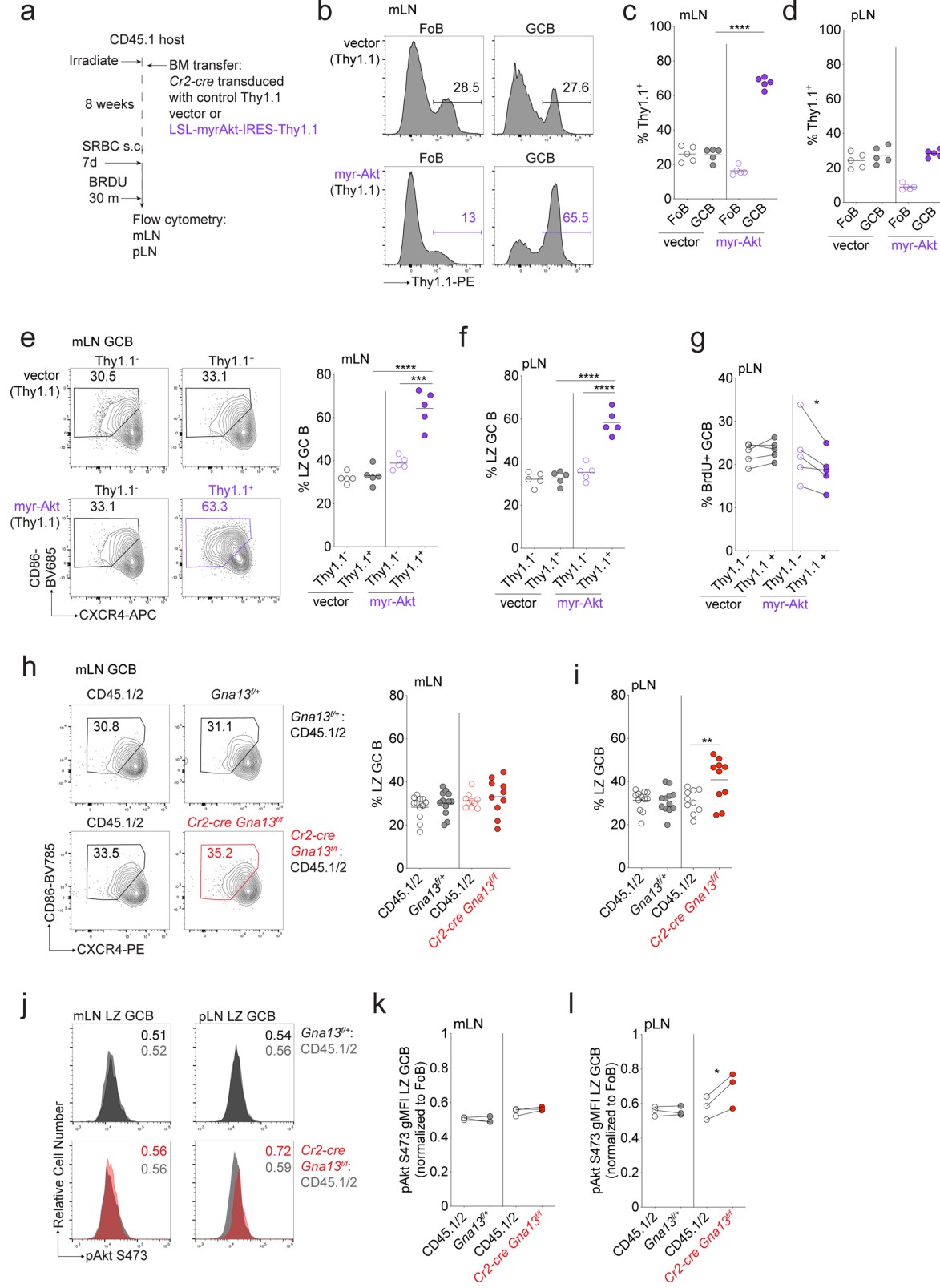

**Extended Data Fig. 3 | See next page for caption.**

**Extended Data Fig. 3 | Gα13-deficiency does not phenocopy high Akt states in mLN GCs. (a)** Experimental scheme for data in b-g. **(b)** Representative flow cytometry plots of Thy1.1 retroviral reporter expression in FoB and GCB from mLN of myr-Akt transduced chimeras. **(c, d)** Percentages of transduced (Thy1.1⁺) FoB and GCB in mLN (c) or pLN (d) of myr-Akt transduced BM chimeras. One experiment, n = 5. ****P < 0.0001 unpaired two-tailed Student's *t*-test for data in c. **(e, f)** Percentages of LZ GC B cells or amongst untransduced (Thy1.1⁻) or transduced (Thy1.1⁺) cells in mLN (e) or pLN (f) of myr-Akt chimeras. One experiment, n = 5. ****P < 0.0001, ***P = 0.0002 unpaired two-tailed Student's *t*-test. **(g)** Percentages of proliferating (BrdU⁺) GC B cells amongst untransduced (Thy1.1⁻) or transduced (Thy1.1⁺) cells in pLN of myr-Akt chimeras. One experiment, n = 5. *P = 0.0465 paired two-tailed Student's *t*-test. **(h-i)** Percentages of LZ GC B cells amongst WT (CD45.1/2) or WT (*Gna13^(fl/+)*) or Gα13-deficient (*Cr2-cre Gna13^(fl/fl)*) cells in mLN (h) or pLN (i) of mixed BM chimeras generated as in Fig. 1a. 2 experiments, n = 12 and 10. **P = 0.0094 unpaired two-tailed Student's *t*-test for data in i. **(j-l)** Intracellular FACS for pAkt S473 in LZ cells from mLN or pLN (j) or gMFI of pAkt S473 normalized to FoB in LZ GC B cells in mLN (k) or pLN (l) in mixed BM chimeras generated as in Fig. 1d. Three experiments, n = 3. *P = 0.0402 paired two-tailed Student's *t*-test for l.

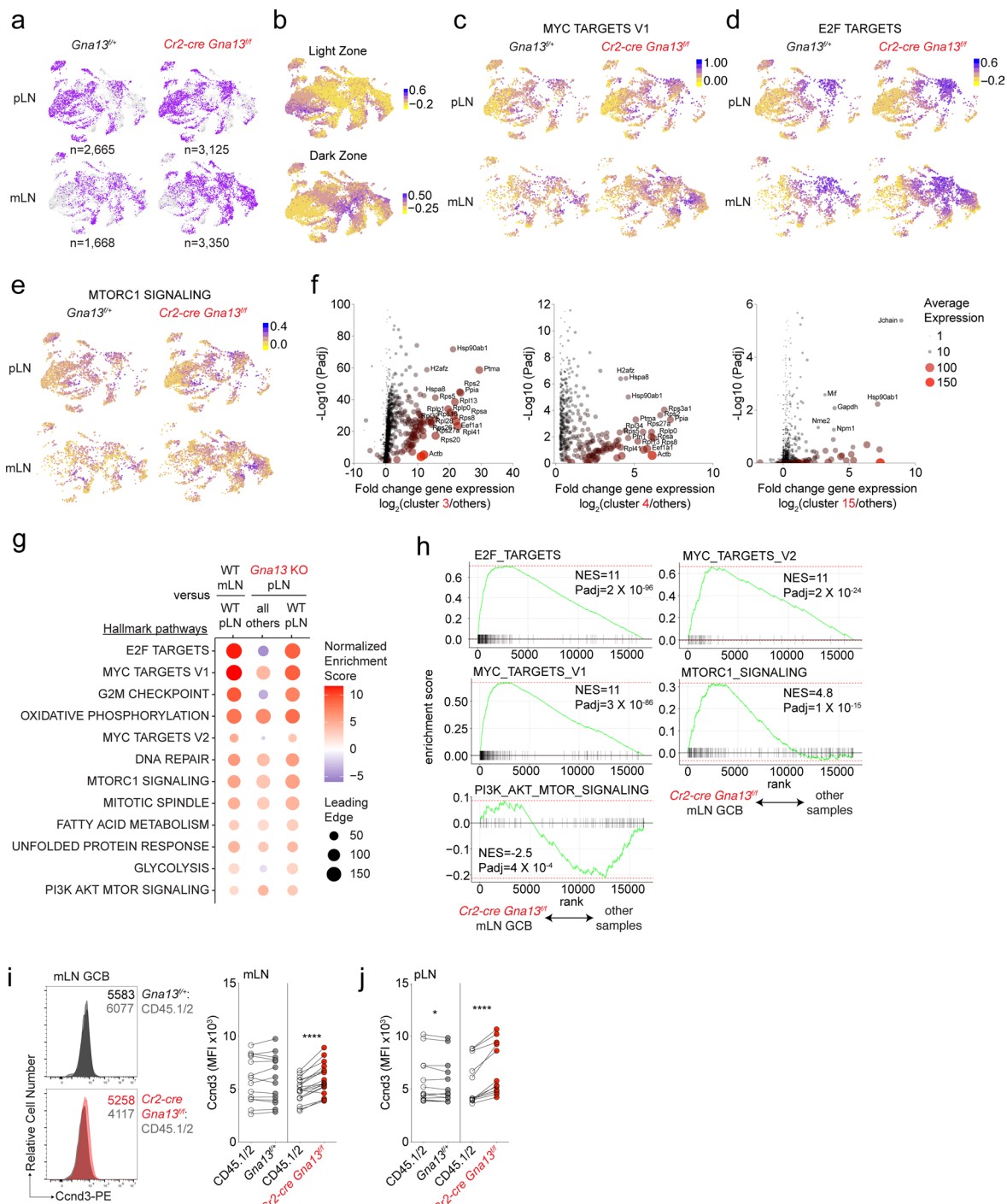

**Extended Data Fig. 4 | Gene signatures enriched in Gα13-deficient mLN GCB.**
**(a)** Distribution of control (*Gna13^(f/+)*) or Gα13-deficient (*Cr2-cre Gna13^(f/f)*) pLN or mLN GCB in UMAP plot. **(b)** Enrichment scores for Light Zone or Dark Zone projected onto UMAP plots. **(c-e)** Enrichment scores for MYC TARGETS V1 (c), E2F TARGETS (d) and MTORC1 SIGNALING (e) gene signatures projected onto UMAP plots of each sample type. **(f)** Differentially expressed genes in clusters with increased representation in Gα13-deficient mLN GC B cells. **(g)** Hallmark

gene set enrichment in WT mLN compared to pLN GC B cells or Gα13-deficient pLN GC B cells compared to all other samples or WT pLN GCB. **(h)** Gene set enrichment plots for Gα13-deficient mLN GCB compared to all other samples. **(i-j)** Intracellular FACS for Cyclin D3 in mLN (i) or pLN (j) from control or Gα13-deficient mixed BM chimeras. Example FACS plots are shown on the left in i with Cyclin D3 MFI indicated. Six experiments, n = 14 and 18 for i. Five experiments, n = 13 and 12 for j. ****P < 0.0001, *P = 0.0326 unpaired two-tailed Student's *t*-test.

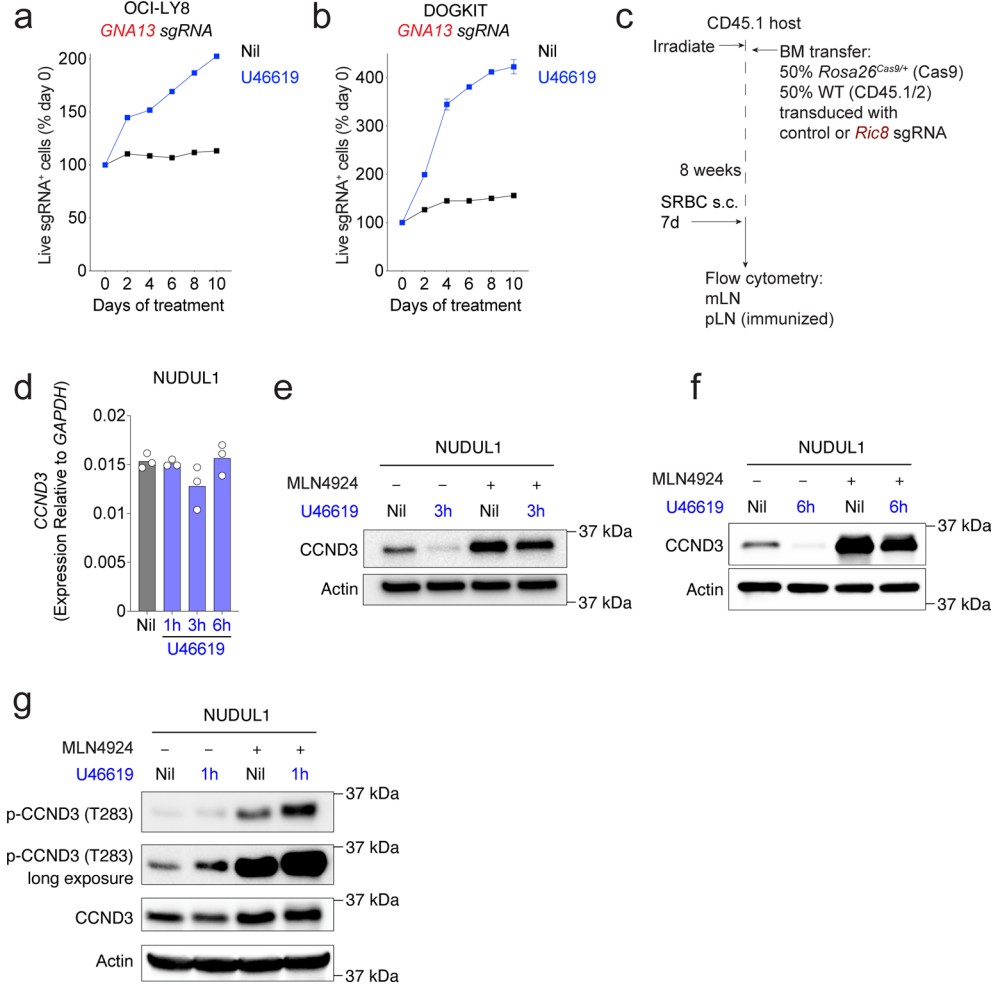

**Extended Data Fig. 5 | Gα13 signaling promotes Cyclin D3 degradation.**
**(a, b)** Relative frequency of *GNA13* sgRNA-expressing OCI-LY8 (a) or Dogkit (b) cells with or without U46619. One experiment representative of 2.
**(c)** Experimental scheme for data in 4 l. **(d)** Quantitative PCR of *CCND3* in NUDUL1 cells treated with U46619. Each point represents a technical replicate.

One experiment representative of 2. **(e-f)** Cyclin D3 protein expression in NUDUL1 cells treated with U46619 and MLN4924 for 3 hours (e) or 6 hours (f). One experiment representative of three or two, respectively. **(g)** Phospho-Cyclin D3 expression in NUDUL1 cells treated with U46619 and MLN4924 for 1 hour. One experiment representative of 2.

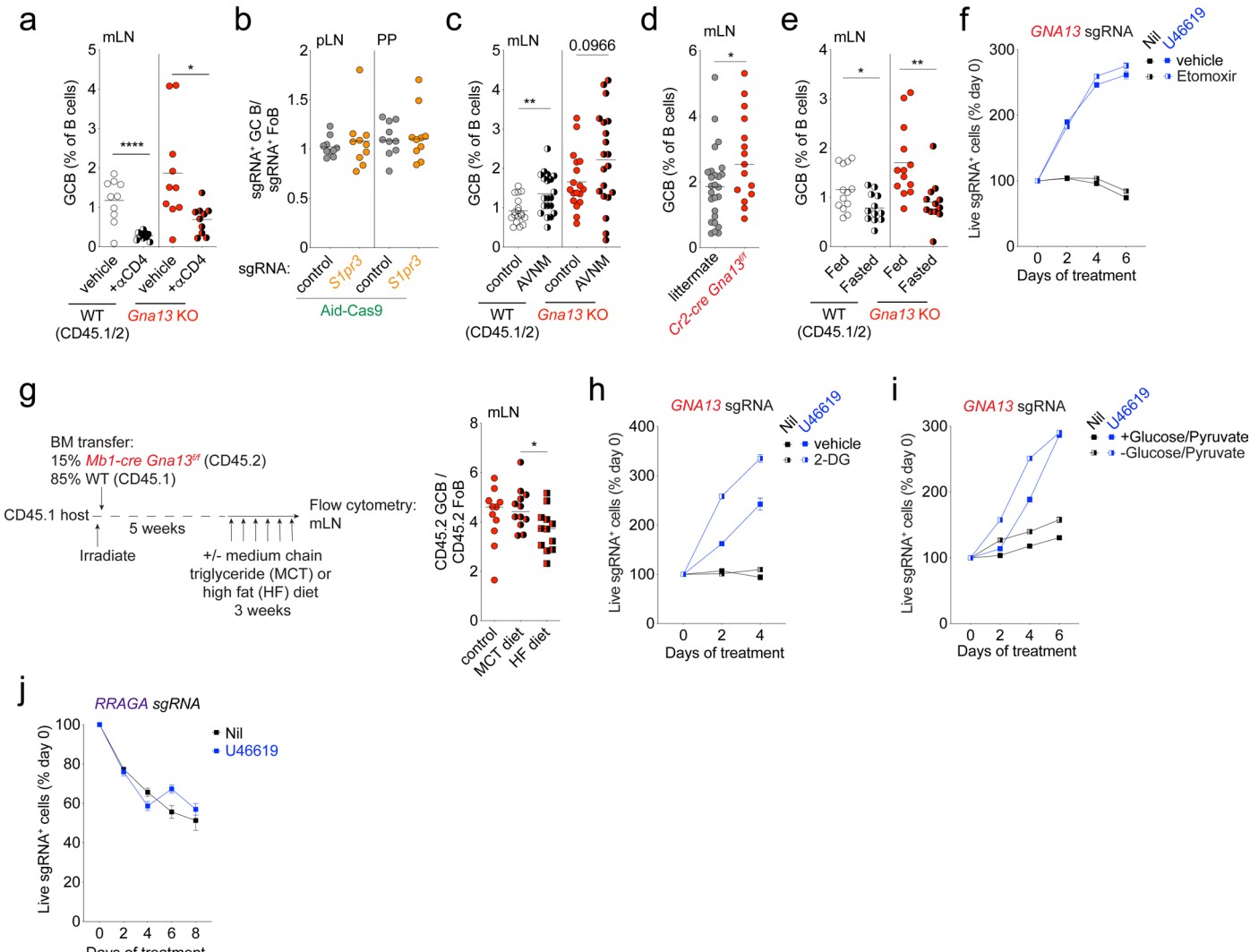

**Extended Data Fig. 6 | Fatty acids, glucose and Rag-dependent nutrients do not differentially support expansion of Gα13-deficient mLN GC B cells.**
**(a)** Frequency of WT or Gα13-deficient GCB as a percentage of B cells in mLN of Gα13-deficient mixed chimeras described in 6e. ****$P < 0.0001$ and *$P = 0.0134$ unpaired two-tailed Student's $t$-test. **(b)** Ratio of sgRNA⁺ GCB to sgRNA⁺ FoB in pLN or PP of Aid-Cas9 BM chimeras targeting *S1pr3*. 2 experiments, n = 10. **(c)** Frequency of WT or Gα13-deficient GCB as a percentage of B cells in mLN of Gα13-deficient mixed chimeras described in 6 h. **$P = 0.0045$ unpaired two-tailed Student's $t$-test. **(d)** Frequency of GCB of B cells in mLN in germ-free *Cr2-cre Gna13^{f/f}* or littermate control (*Gna13^{f/+}*, *Gna13^{f/f}* or *Cr2-cre Gna13^{f/+}*) animals. *$P = 0.0328$ **(e)** Frequency of WT or Gα13-deficient GCB as a percentage of B cells in mLN of Gα13-deficient mixed chimeras described in 6k. *$P = 0.0212$,

**$P = 0.0035$ unpaired two-tailed Student's $t$-test. **(f)** Relative frequency of *GNA13* sgRNA-expressing NUDUL1 cells with U46619 in the presence or absence of the Cpt1 inhibitor, Etomoxir. One experiment representative of 2. **(g)** Ratio of CD45.2 GCB to CD45.2 FoB in mLN of Gα13-deficient mixed chimeras that were fed medium chain triglyceride (MCT) or high fat (HF) diet for 3 weeks prior to analysis. 2 experiments, n = 11, 12 and 12. *$P = 0.0189$ unpaired two-tailed Student's $t$-test. **(h-i)** Relative frequency of *GNA13* sgRNA-expressing NUDUL1 cells with U46619 in the presence of 2-deoxy-glucose (2-DG) (h) or in the absence of glucose and pyruvate (i). One experiment representative of 2. **(j)** Relative frequency of *RRAGA* sgRNA-expressing NUDUL1 cells in the presence or absence of U46619. One experiment representative of three.

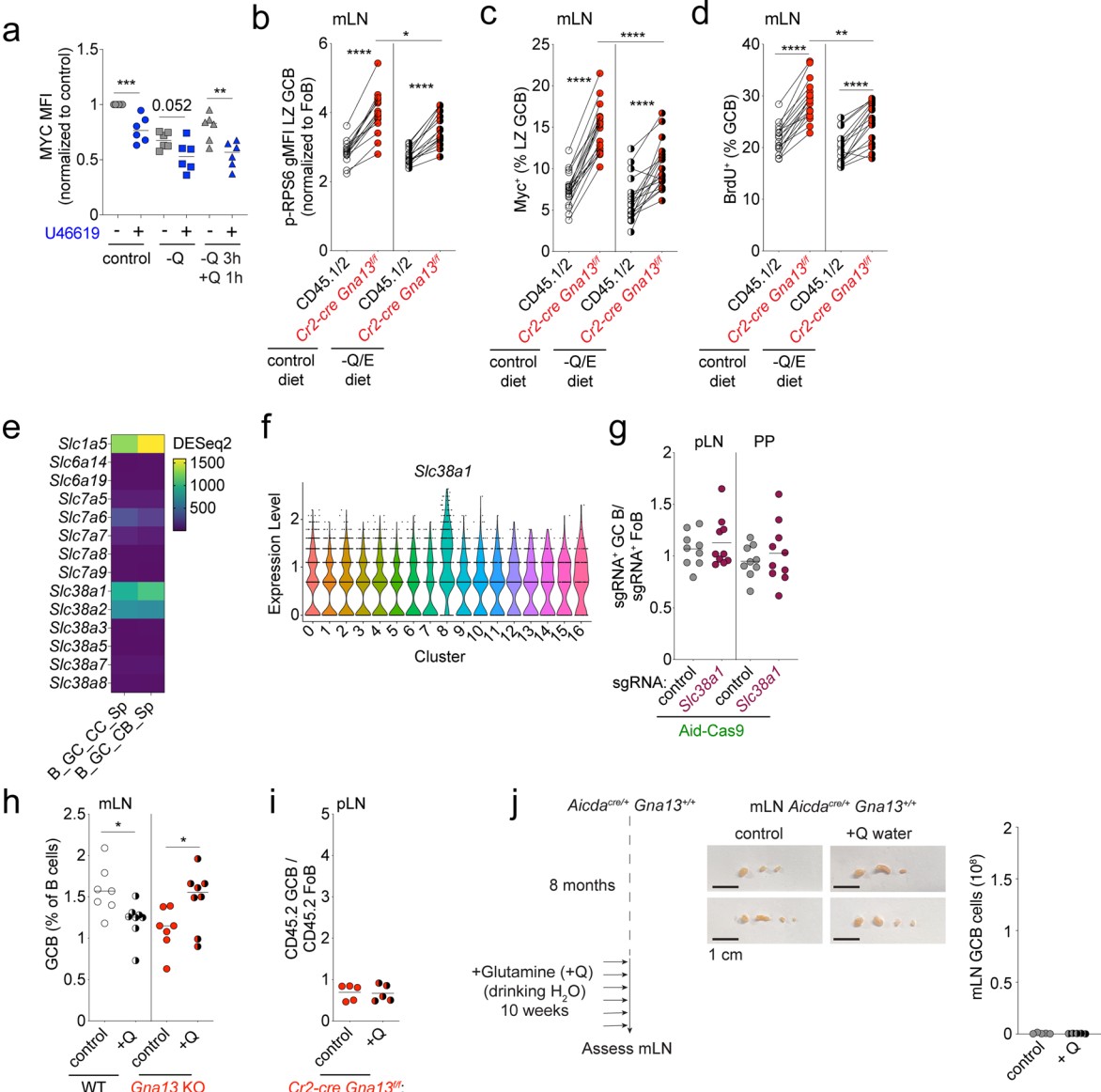

**Extended Data Fig. 7 | Dietary glutamine differentially supports Gα13-deficient mLN GC B cells. (a)** MYC expression in NUDUL1 cells cultured in the absence of glutamine for 4 hours or in the absence of glutamine for 3 hours with glutamine add back for 1 hour prior to analysis in the presence or absence of U46619. Four experiments. ***$P = 0.0009$ and **$P = 0.0062$ unpaired two-tailed Student's *t*-test. Some of the data points from cells not treated with U46619 are the same as in Fig. 7c and are included for comparison. **(b–d)** gMFI of pRPS6 (b) in LZ GCB or percentage of Myc⁺ cells (c) in LZ GCB or percentages of BrdU⁺ cells (d) in GCB in mLN of mixed chimeras that were fed a diet deficient in glutamine and glutamic acid (-Q/E diet) for at least 2 days. Pooled from 7 experiments, n = 15 in b, n = 19 in c. n = 17 and 16 in d. ****$P < 0.0001$ paired two-tailed Student's *t*-test for data within chimeras. *$P = 0.0331$, ****$P < 0.0001$, **$P = 0.0039$ unpaired two-

tailed Student's *t*-test for data between chimeras. **(e)** Expression of glutamine transporters in publicly available RNA sequencing data of sorted GCB (Immgen. org). **(f)** Expression of *Slc38a1* across clusters of GCB from scRNA-seq data described in Fig. 3a. **(g)** Ratio of sgRNA⁺ GC B to sgRNA⁺ FoB in pLN and PP of Aid-Cas9 BM chimeras targeting *Slc38a1*. 2 experiments, n = 9 and 10. **(h-i)** Frequency of WT or Gα13-deficient GCB as a percentage of B cells in mLN (h) or ratio of CD45.2 GCB to CD45.2 FoB in pLN (i) of Gα13-deficient mixed chimeras described in Fig. 7f. *$P = 0.0168$ between WT (CD45.1/2) cells and *$P = 0.0445$ between *Gna13* KO cells unpaired two-tailed Student's *t*-test in h. **(j)** Images of mLN (middle panel), GCB number (right panel) in Gα13-sufficient animals that were given glutamine (28 g/L) in drinking water at 8 months of age for 10 weeks prior to analysis. Scale bars: 1 cm. One experiment, n = 5.

# Reporting Summary

## Statistics

For all statistical analyses, confirm that the following items are present in the figure legend, table legend, main text, or Methods section.

| n/a | Confirmed | |
|---|---|---|
| ☐ | ☒ | The exact sample size (*n*) for each experimental group/condition, given as a discrete number and unit of measurement |
| ☐ | ☒ | A statement on whether measurements were taken from distinct samples or whether the same sample was measured repeatedly |
| ☐ | ☒ | The statistical test(s) used AND whether they are one- or two-sided<br>*Only common tests should be described solely by name; describe more complex techniques in the Methods section.* |
| ☒ | ☐ | A description of all covariates tested |
| ☐ | ☒ | A description of any assumptions or corrections, such as tests of normality and adjustment for multiple comparisons |
| ☐ | ☒ | A full description of the statistical parameters including central tendency (e.g. means) or other basic estimates (e.g. regression coefficient) AND variation (e.g. standard deviation) or associated estimates of uncertainty (e.g. confidence intervals) |
| ☐ | ☒ | For null hypothesis testing, the test statistic (e.g. *F*, *t*, *r*) with confidence intervals, effect sizes, degrees of freedom and *P* value noted<br>*Give P values as exact values whenever suitable.* |
| ☒ | ☐ | For Bayesian analysis, information on the choice of priors and Markov chain Monte Carlo settings |
| ☒ | ☐ | For hierarchical and complex designs, identification of the appropriate level for tests and full reporting of outcomes |
| ☒ | ☐ | Estimates of effect sizes (e.g. Cohen's *d*, Pearson's *r*), indicating how they were calculated |

*Our web collection on statistics for biologists contains articles on many of the points above.*

## Software and code

Policy information about availability of computer code

| | |
|---|---|
| Data collection | Flow cytometry data were collected with CytExpert v2.3.<br>IF data was collected with LASX v4.5<br>IHC and H&E data were collected with Zen v3.8.<br>Western blot images were collected with Image Lab Touch Software v3.01 (BioRad).<br>CRISPR screen reads were generated using NextSeq 1000/2000 Control Software (v. 1.2.036376) (Illumina) and demultiplexed using DRAGEN (v.3.7.4) (Illumina). |
| Data analysis | Flow cytometry data were  analyzed using FlowJo v10.<br>Imaging data were processed and analyzed using Imaris v10 and FlowJo v10.<br>Single cell sequencing files were processed and aligned to mm10, and count matrices were generated using Cell Ranger (v6.0.0). Further analyses were performed in R using the Seurat package (v4). Gene set enrichment analysis was done using the FGSEA workflow.<br>Data analysis was performed on Microsoft Excel v16.80 and Prism v10.<br>GSEA software (v4.0.3)<br>Western blot images were analyzed with Image Lab v6.1 (Bio-Rad).<br>CRISPR screen reads were extracted from fastq files and normalized using custom perl scripts and Bowtie2 (version 2.2.9). |

For manuscripts utilizing custom algorithms or software that are central to the research but not yet described in published literature, software must be made available to editors and reviewers. We strongly encourage code deposition in a community repository (e.g. GitHub). See the Nature Portfolio guidelines for submitting code & software for further information.

## Data

Policy information about availability of data

All manuscripts must include a data availability statement. This statement should provide the following information, where applicable:
  - Accession codes, unique identifiers, or web links for publicly available datasets
  - A description of any restrictions on data availability
  - For clinical datasets or third party data, please ensure that the statement adheres to our policy

> Cell line and single cell RNA seq data was uploaded to the Gene Expression Omnibus (GEO) under accession number GSE253435.
> Repertoire sequencing data and CRISPR screen score (CSS) data used in this study are provided in Figshare (https://figshare.com/s/dc9ea731f2c72b19b4f3)
> Source data are provided with this paper.

## Research involving human participants, their data, or biological material

Policy information about studies with human participants or human data. See also policy information about sex, gender (identity/presentation), and sexual orientation and race, ethnicity and racism.

| | |
|---|---|
| Reporting on sex and gender | Not applicable |
| Reporting on race, ethnicity, or other socially relevant groupings | Not applicable |
| Population characteristics | Not applicable |
| Recruitment | Not applicable |
| Ethics oversight | Not applicable |

Note that full information on the approval of the study protocol must also be provided in the manuscript.

# Field-specific reporting

Please select the one below that is the best fit for your research. If you are not sure, read the appropriate sections before making your selection.

☒ Life sciences          ☐ Behavioural & social sciences          ☐ Ecological, evolutionary & environmental sciences

For a reference copy of the document with all sections, see nature.com/documents/nr-reporting-summary-flat.pdf

# Life sciences study design

All studies must disclose on these points even when the disclosure is negative.

| | |
|---|---|
| Sample size | No statistical methods were used to pre-determine sample sizes, but our sample sizes are similar to those reported in previous publicationsn (Refs. 6, 9, 14, 53). Sample size was based on previous experiments and were of sufficient size to detect differences between groups and were guided by the 3R principles to reduce animal numbers. |
| Data exclusions | No data were excluded from analysis. |
| Replication | Biological and technical replicates were performed. Most of the experiments have been reproduced 2-3 independent times with comparable results. In some instances, data were pooled from independent experiments. Information on replicates is included in figure legends. |
| Randomization | Groups were allocated on the basis of genotype. Littermates that were also sex-matched were used whenever possible to control for co-variates. For experiments with treatment, mice were allocated to control and experimental groups randomly. |
| Blinding | Investigators were not blinded to mouse genotypes or sample type during data collection. Blinding was not feasible due to the staffing requirements that would be needed for blinded experiment and the need for cage labeling in the colony. Data reported are based primarily on quantitative flow cytometry and image quantification analysis which are not subject to the same potential biases as experiments with more subjective readouts. |

# Reporting for specific materials, systems and methods

We require information from authors about some types of materials, experimental systems and methods used in many studies. Here, indicate whether each material, system or method listed is relevant to your study. If you are not sure if a list item applies to your research, read the appropriate section before selecting a response.

## Materials & experimental systems

| n/a | Involved in the study |
|-----|----------------------|
| ☐ | ☒ Antibodies |
| ☐ | ☒ Eukaryotic cell lines |
| ☒ | ☐ Palaeontology and archaeology |
| ☐ | ☒ Animals and other organisms |
| ☒ | ☐ Clinical data |
| ☒ | ☐ Dual use research of concern |
| ☒ | ☐ Plants |

## Methods

| n/a | Involved in the study |
|-----|----------------------|
| ☒ | ☐ ChIP-seq |
| ☐ | ☒ Flow cytometry |
| ☒ | ☐ MRI-based neuroimaging |

## Antibodies

| | |
|---|---|
| Antibodies used | Antibodies used for Flow cytometry:<br>Fixable Viability Dye eFluor 780 (ebiosciences) 1:1000<br>BV786 or BUV395–conjugated anti-B220 (RA3-6B2; BD), 1:400<br>BUV395-conjugated anti-CD4 (RM4-5; BD), 1:400<br>Pacific blue or Alexa Fluor 647–conjugated GL7 (GL-7; BioLegend), 1:400<br>BV650 conjugated anti-IgD (11- 26c.2a; BioLegend), 1:400<br>PerCP Cy5.5 or PE-Cy7–conjugated anti-CD38 (90; BioLegend), 1:400<br>PE-Cy7, PE or BV421-conjugated anti-Fas (Jo2; BD), 1:400<br>FITC, PerCP-Cy5.5 or Alexa Flour 700-conjugated anti-CD45.2 (104; BioLegend), 1:400<br>PE-Cy7 or PerCP-Cy5.5–conjugated anti-CD45.1 (A20; BioLegend), 1:400<br>BV786–conjugated anti-CD86 (GL-1; BioLegend), 1:100<br>PE-conjugated, APC- conjugated anti-CXCR4 (2B11; ebiosciences), 1:100<br>Alexa Fluor 647–conjugated anti-c-Myc (Y69; Abcam), 1:400<br>PE-conjugated anti-Cyclin D3 (DCS-22; Biolegened), 1:400<br>FITC or APC conjugated anti-BrdU (BU20A; eBioscience), 1:100<br>p-AKT S473 (D9E; Cell Signaling; 1:400)<br>pRPS6  Ser240/244  (D68F8; Cell Signaling) 1:400<br>AF647-conjugated anti-rabbit IgG (Invitrogen)1:1000<br><br>For Immunohistochemistry experiments:<br>biotinylated anti-GL7 (GL7; Biolegend), 1:400<br>biotinylated anti-B220 (RA3-6B2; Biolegend), 1:200<br>biotinylated anti-CD35 (8C12; BD Biosciences), 1:200<br>unlabeled polyclonal goat anti-mouse IgD (Cedarlane) 1:1000<br><br>For Immunofluorescence experiments<br>SparkRed 718-conjugated anti-B220 (RA3-6B2; BioLegend) 1:200<br>AF488-conjugated anti-IgD (11-26c.2a; BioLegend) 1:100<br>AF647–conjugated GL7 (GL-7; BioLegend) 1:100<br>biotin-conjugated anti-CD35 (8C12; BD) 1:200<br>biotin-conjugated anti-LYVE1 (ALY7, eBioscience) 1:200<br>rabbit anti-Myc (D3N8F; Cell Signaling) 1:400<br>rabbit anti-phospho-ribosomal protein S6 (pRPS6) (Ser240/244) (D68F8; Cell Signaling) 1:800<br>BV421-conjugated streptavidin (BD) 1:400<br>AF555-conjugated anti-rabbit IgG (Invitrogen)1:200<br>JOPRO-1 (Invitrogen) 1:10,000<br><br>For Western blot experiments:<br>MYC (D3N8F; Cell Signaling; 1:2000)<br>CCND3 (DCS22; Cell Signaling; 1:2000)<br>Actin (13E5; Cell Signaling; 1:4000)<br>p-P70S6K T389 (108D2; Cell Signaling; 1:2000)<br>P70S6K (polyclonal rabbit antibody #9202; Cell Signaling; 1:2000)<br>p-RPS6 S235/6 (D57.2.2E; Cell Signaling; 1:4000)<br>RPS6 (5G10; Cell Signaling; 1:4000)<br>p-AKT S473 (D9E; Cell Signaling; 1:4000)<br>p-AKT T308 (D25E6; Cell Signaling; 1:4000)<br>p-CCND3 T283 (E1V6W; Cell Signaling; 1:2000) |
| Validation | All antibodies used in this study are from widely used commercial sources and have been previously described in the literature and validated by the vendors. Validation data and citation information are available on the manufacturer's website (links below). Appropriate antibody dilutions were performed based on preliminary experiments and intensity of fluorescent signals. Dilutions for flow cytometry antibodies are referred to a staining volume of 40 ul per sample (~1 × 106 cells).<br><br>Antibodies used for Flow cytometry:<br>Fixable Viability Dye eFluor 780 (ebiosciences) 1:1000<br>https://www.thermofisher.com/order/catalog/product/65-0865-14 |

BV786 or BUV395–conjugated anti-B220 (RA3-6B2; BD), 1:400
https://www.bdbiosciences.com/en-us/products/reagents/flow-cytometry-reagents/research-reagents/single-color-antibodies-ruo/
buv395-rat-anti-mouse-cd45r-b220.563793
https://www.bdbiosciences.com/en-us/products/reagents/flow-cytometry-reagents/research-reagents/single-color-antibodies-ruo/
bv786-rat-anti-mouse-cd45r-b220.563894

BUV395-conjugated anti-CD4 (RM4-5; BD), 1:400
https://www.bdbiosciences.com/en-us/products/reagents/flow-cytometry-reagents/research-reagents/single-color-antibodies-ruo/
buv395-rat-anti-mouse-cd4.568375

Pacific blue or Alexa Fluor 647–conjugated GL7 (GL-7; BioLegend), 1:400
https://www.biolegend.com/en-gb/products/pacific-blue-anti-mouse-human-gl7-antigen-t-and-b-cell-activation-marker-
antibody-9580
https://www.biolegend.com/en-gb/products/alexa-fluor-647-anti-mouse-human-gl7-antigen-t-and-b-cell-activation-marker-
antibody-8602

BV650 conjugated anti-IgD (11- 26c.2a; BioLegend), 1:400
https://www.biolegend.com/en-gb/products/brilliant-violet-650-anti-mouse-igd-9031

PerCP Cy5.5 or PE-Cy7–conjugated anti-CD38 (90; BioLegend), 1:400
https://www.biolegend.com/en-gb/products/pe-cyanine7-anti-mouse-cd38-antibody-3926
https://www.biolegend.com/en-gb/products/percp-cyanine5-5-anti-mouse-cd38-antibody-9563

PE-Cy7, PE or BV421-conjugated anti-Fas (Jo2; BD), 1:400
https://www.bdbiosciences.com/en-us/products/reagents/flow-cytometry-reagents/research-reagents/single-color-antibodies-ruo/
bv421-hamster-anti-mouse-cd95.562633
https://www.bdbiosciences.com/en-us/products/reagents/flow-cytometry-reagents/research-reagents/single-color-antibodies-ruo/
pe-cy-7-hamster-anti-mouse-cd95.557653
https://www.bdbiosciences.com/en-us/products/reagents/flow-cytometry-reagents/research-reagents/single-color-antibodies-ruo/
pe-hamster-anti-mouse-cd95.554258

FITC, PerCP-Cy5.5 or Alexa Flour 700-conjugated anti-CD45.2 (104; BioLegend), 1:400
https://www.biolegend.com/en-gb/products/fitc-anti-mouse-cd45-2-antibody-6
https://www.biolegend.com/en-gb/products/percp-cyanine5-5-anti-mouse-cd452-antibody-4271
https://www.biolegend.com/en-gb/products/alexa-fluor-700-anti-mouse-cd45-2-antibody-3393

PE-Cy7 or PerCP-Cy5.5–conjugated anti-CD45.1 (A20; BioLegend), 1:400
https://www.biolegend.com/en-gb/products/pe-cyanine7-anti-mouse-cd45-1-antibody-4917
https://www.biolegend.com/en-gb/products/percp-cyanine5-5-anti-mouse-cd45-1-antibody-4269

BV786–conjugated anti-CD86 (GL-1; BioLegend), 1:100
https://www.biolegend.com/en-gb/products/brilliant-violet-785-anti-mouse-cd86-antibody-12818

PE-conjugated, APC- conjugated anti-CXCR4 (2B11; ebiosciences), 1:100
https://www.thermofisher.com/antibody/product/CD184-CXCR4-Antibody-clone-2B11-Monoclonal/12-9991-82
https://www.thermofisher.com/antibody/product/CD184-CXCR4-Antibody-clone-2B11-Monoclonal/17-9991-82?imageId=1023016

Alexa Fluor 647–conjugated anti-c-Myc (Y69; Abcam), 1:400
https://www.abcam.com/products/primary-antibodies/alexa-fluor-647-c-myc-antibody-y69-ab190560.html

PE-conjugated anti-Cyclin D3 (DCS-22; Biolegend), 1:400
https://www.biolegend.com/en-gb/products/pe-anti-cyclin-d3-antibody-14405

FITC or APC conjugated anti-BrdU (BU20A; eBioscience), 1:100
https://www.thermofisher.com/antibody/product/BrdU-Antibody-clone-BU20A-Monoclonal/11-5071-42
https://www.thermofisher.com/antibody/product/BrdU-Antibody-clone-BU20A-Monoclonal/17-5071-42

p-AKT S473 (D9E; Cell Signaling; 1:400)
https://www.cellsignal.com/products/primary-antibodies/phospho-akt-ser473-d9e-xp-rabbit-mab/4060

pRPS6  Ser240/244  (D68F8; Cell Signaling) 1:400
https://www.cellsignal.com/products/primary-antibodies/phospho-s6-ribosomal-protein-ser240-244-d68f8-xp-rabbit-mab/5364

AF647-conjugated anti-rabbit IgG (Invitrogen)1:1000
https://www.thermofisher.com/antibody/product/Goat-anti-Rabbit-IgG-H-L-Highly-Cross-Adsorbed-Secondary-Antibody-Polyclonal/
A-21245

For Immunohistochemistry experiments:
biotinylated anti-GL7 (GL7; Biolegend), 1:400
https://www.biolegend.com/en-gb/products/biotin-anti-mousehuman-gl7-antigen-t-and-b-cell-activation-marker-antibody-15161

biotinylated anti-B220 (RA3-6B2; Biolegend), 1:200
https://www.biolegend.com/en-gb/products/biotin-anti-mouse-human-cd45r-b220-antibody-444

biotinylated anti-CD35 (8C12; BD Biosciences), 1:200
https://www.bdbiosciences.com/en-us/products/reagents/flow-cytometry-reagents/research-reagents/single-color-antibodies-ruo/

biotin-rat-anti-mouse-cd35.553816

unlabeled polyclonal goat anti-mouse IgD (Cedarlane) 1:1000
https://www.cedarlanelabs.com

For Immunofluorescence experiments
SparkRed 718-conjugated anti-B220 (RA3-6B2; BioLegend) 1:200
https://www.biolegend.com/en-gb/products/spark-red-718-anti-mouse-human-cd45r-b220-antibody-22290

AF488-conjugated anti-IgD (11-26c.2a; BioLegend) 1:100
https://www.biolegend.com/en-gb/products/alexa-fluor-488-anti-mouse-igd-7092

AF647–conjugated GL7 (GL-7; BioLegend) 1:100
https://www.biolegend.com/en-gb/products/alexa-fluor-647-anti-mouse-human-gl7-antigen-t-and-b-cell-activation-marker-antibody-8602

biotin-conjugated anti-CD35 (8C12; BD) 1:200
https://www.bdbiosciences.com/en-us/products/reagents/flow-cytometry-reagents/research-reagents/single-color-antibodies-ruo/biotin-rat-anti-mouse-cd35.553816

biotin-conjugated anti-LYVE1 (ALY7, eBioscience) 1:200
https://www.thermofisher.com/antibody/product/LYVE1-Antibody-clone-ALY7-Monoclonal/13-0443-82

rabbit anti-Myc (D3N8F; Cell Signaling) 1:400
https://www.cellsignal.com/products/primary-antibodies/c-myc-n-myc-d3n8f-rabbit-mab/13987

rabbit anti-phospho-ribosomal protein S6 (pRPS6) (Ser240/244) (D68F8; Cell Signaling) 1:800
https://www.cellsignal.com/products/primary-antibodies/phospho-s6-ribosomal-protein-ser240-244-d68f8-xp-rabbit-mab/5364

BV421-conjugated streptavidin (BD) 1:400
https://www.bdbiosciences.com/en-us/products/reagents/flow-cytometry-reagents/research-reagents/single-color-antibodies-ruo/bv421-streptavidin.563259

AF555-conjugated anti-rabbit IgG (Invitrogen)1:200
https://www.thermofisher.com/antibody/product/Goat-anti-Rabbit-IgG-H-L-Cross-Adsorbed-Secondary-Antibody-Polyclonal/A-21428

JOPRO-1 (Invitrogen) 1:10,000
https://www.thermofisher.com/order/catalog/product/Y3603

For Western blot experiments:
MYC (D3N8F; Cell Signaling; 1:2000)
https://www.cellsignal.com/products/primary-antibodies/c-myc-n-myc-d3n8f-rabbit-mab/13987

CCND3 (DCS22; Cell Signaling; 1:2000)
https://www.cellsignal.com/products/primary-antibodies/cyclin-d3-dcs22-mouse-mab/2936

Actin (13E5; Cell Signaling; 1:4000)
https://www.cellsignal.com/products/primary-antibodies/b-actin-13e5-rabbit-mab/4970

p-P70S6K T389 (108D2; Cell Signaling; 1:2000)
https://www.cellsignal.com/products/primary-antibodies/phospho-p70-s6-kinase-thr389-108d2-rabbit-mab/9234

P70S6K (polyclonal rabbit antibody #9202; Cell Signaling; 1:2000)
https://www.cellsignal.com/products/primary-antibodies/p70-s6-kinase-antibody/9202

p-RPS6 S235/6 (D57.2.2E; Cell Signaling; 1:4000)
https://www.cellsignal.com/products/primary-antibodies/phospho-s6-ribosomal-protein-ser235-236-d57-2-2e-xp-rabbit-mab/4858

RPS6 (5G10; Cell Signaling; 1:4000)
https://www.cellsignal.com/products/primary-antibodies/s6-ribosomal-protein-5g10-rabbit-mab/2217

p-AKT S473 (D9E; Cell Signaling; 1:4000)
https://www.cellsignal.com/products/primary-antibodies/phospho-akt-ser473-d9e-xp-rabbit-mab/4060

p-AKT T308 (D25E6; Cell Signaling; 1:4000)
https://www.cellsignal.com/products/primary-antibodies/phospho-akt-thr308-d25e6-xp-rabbit-mab/13038

p-CCND3 T283 (E1V6W; Cell Signaling; 1:2000)
https://www.cellsignal.com/products/primary-antibodies/phospho-cyclin-d3-thr283-e1v6w-rabbit-mab/53966

# Eukaryotic cell lines

Policy information about cell lines and Sex and Gender in Research

| | |
|---|---|
| Cell line source(s) | NUDUL1 Human diffuse large B cell line ATCC CRL-2969<br>OCI-Ly8 Human diffuse large B cell line NCI<br>Dogkit Human diffuse large B cell line DMSZ<br>The Platinum E (Plat-E ) retroviral packaging cell line was a gift from Susan R. Schwab at New York University.<br>293FT Transformed Human kidney cell line Thermo Fisher R70007 |
| Authentication | We used a "DNA fingerprinting" method to test for the presence or absence of 16 common copy number variants allowing the detection of cross-contamination. |
| Mycoplasma contamination | All cell line tested negative for mycoplasma using using the MycoAlert Mycoplasma Detection Kit (Lonza). Cell lines were tested regularly and preventative treatment was undertaken using MycoZap (Lonza) and Plasmocin (InvivoGen). |
| Commonly misidentified lines<br>(See ICLAC register) | No cell lines used in this study among commonly misidentified lines. |

# Animals and other research organisms

Policy information about studies involving animals; ARRIVE guidelines recommended for reporting animal research, and Sex and Gender in Research

| | |
|---|---|
| Laboratory animals | Male and female mus musculus were used.<br>In bone marrow chimeras experiments, animals were irradiated between 6 and 10 weeks of age and analyzed 7-10 weeks following irradiation (Fig. 1e,f,g, Fig. 2, Fig. 3e-g, j-l, Fig. 4l, Fig. 6b-e, g-h, k-l, Fig. 7d-f, ED Fig. 2b-f, ED Fig. 3, ED Fig. 4i-j, ED Fig. 6a-c, e,g, ED Fig. 7b-d, g h-i) .  Most other experiments were performed on adult animals between 7 and 20 weeks of age (Fig. 1d, h-l, Fig. 3a-d, h, m, Fig. 6f, i-j, ED Fig. 2a, f, g, ED Fig. 4a-h, ED Fig. 6d). Aging cohorts of Galpha13 deficient and littermate control mice were analyzed between 10 and 25 months of age (Fig. 1a-c and ED Fig. 1). Some glutamine supplementation experiments were started on mice that were 8 months old and analyzed after 10 weeks of treatment (Fig. 7g and ED Fig. 7j). All mice were on a C57BL/6 background. Mice were housed in a specific pathogen–free environment (except in gnotobiotic experiments) in ventilated microisolator cages with 12 h light and 12 h dark cycles at 72 F and 40–60% relative humidity. All mouse experiments received approval by the National Cancer Institute Animal Care and Use Committee (NCI-ACUC) and were performed in accordance with NCI-ACUC guidelines and under approved protocol LYMB-001.<br><br>Adult B6-Ly5.1/Cr (B6.SJL-PtprcaPepcb/BoyCrCrl; Stock number 564) mice at least 6 weeks of age were from Charles River Frederick Research Model Facility. Cr2-cre (B6.CgTg(Cr2-cre)3Cgn/J; Stock number 006368), Aicdacre (B6.129P2-Aicdatm1(cre)Mnz/J; Stock number 007770), Rosa26LSL-Cas9 (B6J.129(B6N)-Gt(ROSA)26Sortm1(CAG-cas9*,-EGFP)Fezh/J; Stock number 026175), Rosa26Cas9 (B6(C)-Gt(ROSA)26Sorem1.1(CAG-cas9*,-EGFP)Rsky/J; Stock number 028555) and Mb1-cre (B6.C(Cg)-Cd79atm1(cre)Reth/EhobJ; Stock number 020505) mice were from The Jackson Laboratory. Gna13f/f mice were from S. Coughlin (University of California, San Francisco, San Francisco, CA). S1pr2-creERT2 BAC-transgenic and Rosa26LSL-tdTomato were from T. Okada (RIKEN, Yokohama City, Kanagawa, Japan) |
| Wild animals | This study did not utilize wild animals. |
| Reporting on sex | Male and female mice were used in this study. Littermates that were also sex-matched were used whenever possible to control for co-variates. |
| Field-collected samples | This study did not utilize field-collected samples. |
| Ethics oversight | National Cancer Institute Animal Care and Use Committee (NCI-ACUC). Protocol number: LYMB-001 |

Note that full information on the approval of the study protocol must also be provided in the manuscript.

# Plants

| | |
|---|---|
| Seed stocks | Not applicable |
| Novel plant genotypes | Not applicable |
| Authentication | Not applicable |

# Flow Cytometry

## Plots

Confirm that:

☒ The axis labels state the marker and fluorochrome used (e.g. CD4-FITC).

☒ The axis scales are clearly visible. Include numbers along axes only for bottom left plot of group (a 'group' is an analysis of identical markers).

☒ All plots are contour plots with outliers or pseudocolor plots.

☒ A numerical value for number of cells or percentage (with statistics) is provided.

## Methodology

| | |
|---|---|
| Sample preparation | Mesenteric lymph node, peripheral lymph node or peyer's patch cell suspensions were generated by mashing the organs through 70-mm cell strainers in RPMI containing 2% (v/v) FBS, antibiotics (penicillin (50 IU/ml) and streptomycin (50 mg/ml); Cellgro) and 10 mM HEPES, pH 7.2 (Cellgro). |
| Instrument | Beckman Coulter Cytoflex LX for cell analysis. Sony MA900 for cell sorting. |
| Software | Data was collected using CytExpert v2.3 (Beckman Coulter). Data was analyzed with Flowjo v10 (Treestar) |
| Cell population abundance | In all flow cytometry experiments, 300,000 to 1,000,000 events were recorded per sample. For repertoire sequencing, 8,000 to 42,000 GC B cells were sorted into RLT buffer (Qiagen). GC B cells were sorted as single live cells that were B220+, CD4-, IgDlo, CD38lo, Fas+ and GL7+. For single cell RNA sequencing experiments, ~12,000 cells per sample were sorted into PBS + 0.08% BSA. |
| Gating strategy | Relevant examples of gating strategies are shown in the main and supplemental figures. |

☒ Tick this box to confirm that a figure exemplifying the gating strategy is provided in the Supplementary Information.

