## [Peer Review File · Nature Immunology]

Gα13 restricts nutrient driven proliferation in mucosal germinal centers

Corresponding Author: Dr Jagan Muppidi

Version 0:

Decision Letter:

20th Feb 2024

Dear Jagan,

Thank you for providing a point-by-point response to the referees' comments on your manuscript entitled, "Gα13 restricts nutrient driven proliferation in mucosal germinal centers". As noted previously, all three referees find your work of considerable potential interest, but they have raised several concerns that should be addressed. In light of these comments, we cannot accept the current version of manuscript for publication, but would be very interested in considering a revised version that addresses these concerns along the lines proposed in your response. Thus, we invite you to submit a substantially revised manuscript, however please bear in mind that we will be reluctant to approach the referees again in the absence of major revisions.

Specifically, the revision should include new experiments to address:

- (1) additional analysis of the PI3K-AKT-mTOR transcriptional signature to compare pLNs vs mLNs from Ga13-deficient and WT controls.
- (2) examine p-RPS6, Myc and BrdU in GC B cells from pLNs and mLNs from Ga13-deficient/WT mixed chimeric mice. Perform separate experiments with mixed chimeric mice receiving (or not) dietary glutamine supplementation.
- (3) increase the sample size of Ga13-deficient and WT controls examined for the experiment shown in Fig. 7f (mice receiving 10 weeks of glutamine supplementation).
- (4) examine scRNA-seq clusters 3,4,15 that are increased in the Ga13-deficient GC B cell population; provide more annotation for the most-affected gene features and include an analysis of gene signature enrichment across samples.
- (5) perform experiments in cell lines targeting to AKT1, AKT2, AKT3 or PTEN with sgRNA and determine whether stimulation of Ga13 can still suppress mTORC1 signaling and MYC expression.

Please include the additional textual clarifications as indicated in your response letter.

When you revise your manuscript, please take into account all reviewer and editor comments, please highlight all changes in the manuscript text file in Microsoft Word format.

* If you have not done so already please begin to revise your manuscript so that it conforms to our Article format instructions at <http://www.nature.com/ni/authors/index.html>. Refer also to any guidelines provided in this letter.

The Reporting Summary can be found here:

Link Redacted

If you wish to submit a suitably revised manuscript we would hope to receive it within 6 months. If you cannot send it within this time, please let us know. We will be happy to consider your revision so long as nothing similar has been accepted for publication at Nature Immunology or published elsewhere.

Nature Immunology is committed to improving transparency in authorship. As part of our efforts in this direction, we are now requesting that all authors identified as 'corresponding author' on published papers create and link their Open Researcher and Contributor Identifier (ORCID) with their account on the Manuscript Tracking System (MTS), prior to acceptance. ORCID helps the scientific community achieve unambiguous attribution of all scholarly contributions. You can create and link your ORCID from the home page of the MTS by clicking on 'Modify my Springer Nature account'. For more information please visit please visit www.springernature.com/orcid.

Thank you for the opportunity to review your work.

Kind regards,

Laurie

Laurie A. Dempsey, Ph.D.
Senior Editor
Nature Immunology
l.dempsey@us.nature.com
ORCID: 0000-0002-3304-796X

Referee expertise:

Referee #1: Lymphocyte migration

Referee #2: Immune cell metabolism

Referee #3: Germinal center responses

Reviewers' Comments:

Reviewer #1:

Remarks to the Author:

In this manuscript, Nguyen et al. show that loss of Ga13 drives B cell lymphomas preferentially in the mesenteric lymph nodes, and argue that the mesenteric lymph nodes are unique because Ga13 is required to counteract mTORC1 signaling driven by the availability of dietary glutamine. Their study contributes to the very interesting questions of (1) why immune responses differ in differently situated lymph nodes, and (2) what is distinct about the gut microenvironment beyond the

microbiome. The experiments are logical and well-controlled, and my comments are relatively minor.

1) The authors show that strong induction of pAKT by over-expression of myrAkt or loss of PTEN does not phenocopy loss of Ga13 in mLN, although induction of pAKT may explain the effect of Ga13 loss in pLN. For completeness, it would be interesting to assess (for example by flow cytometry) whether pAKT is measurably changed in Ga13-KO GC B cells compared to control in the pLN or mLN.

2) The authors argue that the availability of glutamine in the mLN  mTORC1 activation (upon loss of Ga13)  Myc expression  outgrowth. The evidence for mTORC1 activation in the mLN is in Fig. 3J,K, where the difference in pRPS6 upon loss of Ga13 is subtle. It would be helpful to extend this. Would flow cytometry enable analysis of a larger number of cells? What happens in the pLN? Does Q limitation affect pRPS6? Does Q supplementation activate the pathway (here it would be nice to see Myc and BrdU also)?

3) In Figs. 6C, 6D, and 7D, it would be helpful to see the ratio of % Myc+ (or % BrdU+) among Ga13-KO GC B cells to the % Myc+ (or % BrdU+) among control GC B cells, from mice tx with vehicle and mice tx with rapamycin or dietary restriction. This would help distinguish the global effect of treatment on all GC B cells from effects specific to Ga13-KO GC B cells.

Reviewer #2:

Remarks to the Author:

In this manuscript Nguyen et al. study the role of Galpha13 in mucosal germinal centers (GC). Galpha13 is known to limit GC expansion by inhibiting Akt signaling. Here the authors show that Galpha13 deficiency increases the frequency of cMyc+ cells and mTORC1 activity. Moreover, the authors demonstrate that Galpha13 deficient GC B cells remain more competitive in the absence of gut microbiota or T cell help. Interestingly, Galpha13 deficient GC B cells seem to rely less on fatty acid metabolism than normal GC B cells and instead their expansion is (partially) supported by glutamine. In my opinion the suggested metabolic reprogramming in the absence of Galpha13 is the most interesting part of the study and could play a role in developing new therapies. Unfortunately, one the most interesting experiments (Fig.7F) is underpowered. This experiment should be expanded to include more experimental animals, wildtype controls and mice fed with other nutrients (for example glucose). This would allow to test whether a general increase in high-energy nutrients fastens lymphoma development in this mouse model or whether glutamine plays a special role.

Minor concerns:

- Throughout the manuscript the authors show the percentage of GC B cells in the population of life cells. The authors should add a graph showing the percentage of GC B cells in the population of total B cells.

- In Fig.1K the authors should specify the markers used to define memory B cells in the figure legends

- The authors should specify the genotype of controls used. Where the control mice cre positive? If not, can the authors provide evidence to exclude the contribution of cre recombinase expression to the observed phenotype.

- The authors write " Galpha13 suppresses mLN GC proliferation independently of PI3K/Akt " and "In the absence of Galpha13, loss of Pten promoted GC outgrowths and LZ expansion suggesting that Pten and Galpha13 suppress GC B cell expansion via distinct pathways ". These statements are in my opinion unjustified. The presented data do not exclude that Galpha13 suppresses mLN GC proliferation by activating PI3K/Akt. One could imagine different scenarios in which Galpha13 deletion does not lead to LZ polarization, despite Akt signaling driving GC expansion. For example, it is possible that while Galpha13 loss, Pten loss as well as Akt-hyperactivation all lead to an increase in signaling downstream of Akt, the magnitude of signaling activity may differ. An intermediate increase in Akt signaling in Galpha13-deficient B cells might support survival, but not polarization unlike a stronger Akt signal in Pten deficient or Myr-Akt expressing B cells. The authors should assess activity of the Akt signaling pathway in their models and change the language of the text to avoid misleading statements.

- In Fig.2K the authors should show an example staining for pS6

- The authors write " We found that in the presence of silvestrol and MLN4924, Galpha13 stimulation was not able to further reduce MYC protein (Fig. 5E). Collectively, these data suggest that sustained Galpha13-mediated loss of MYC protein is primarily a result of reduced translation (Fig. 5F). " I think this statement cannot be made. In the shown experiment (Fig.5E) Myc protein levels are already very low when treated with the combination of MLN4924 and silvestrol, the band after U46619 treatment appears to be slightly weaker. The authors might not be able to observe strong effects of U46619 treatment because Myc levels before the treatment are already very low. I don't think this experiment convincingly shows that Galpha13-mediated loss of MYC protein is primarily a result of reduced translation.

Reviewer #3:

Remarks to the Author:

Muppidi JR and colleagues have previously shown that Ga13 deficiency in mature B cells leads to an increase in GC B cells in mLN and the development of GCB-DLBCL. This study investigates the selective impact of Ga13 deficiency in mLN over pLN. Through genetic tools, scRNA-seq, and metabolic assays, the authors suggest that tumorigenesis from Ga13 deficiency primarily involves the mTORC1/Myc pathway, rather than PI3K/AKT. They also present evidence suggesting that

dietary glutamine plays a significant role in regulating GC dynamics in mLNs in the context of Ga13 deficiency. The novelty of this study lies in its mechanistic insights and the proposed role of dietary metabolites, specifically glutamine, in differentially influencing oncogenic pathways in mLNs and pLNs, provided the findings are strongly supported.

Major questions to be addressed

1. The pivotal role of glutamine in preferential tumorigenesis due to Ga13 deficiency in mLNs versus pLNs is unclear. The authors show data on Gln supplementation in drinking water causing lymphadenopathy in mLNs (Fig. 7F), but the study has limitations, including a small sample size and the absence of a control group with Ga13 sufficient mice also receiving Gln supplementation.
2. The study does not confirm whether the combination of Ga13 deficiency and Gln supplementation is enough to induce DLBCL in pLNs, as it does in mLNs. A possible approach would be to increase Gln levels in pLNs through systemic supplementation and observe any increase in lymphadenopathy, specifically under Ga13 deficiency, as a sign of DLBCL development.
3. scRNA-seq provides valuable information for the changes caused by Ga13 deficiency, i.e. clusters 3, 4 and 15 expanded in mLNs and further enhanced by Ga13 deficiency. However, it is somehow disappointing that these clusters have not been thoroughly characterized. They were not annotated, though being excluded from annotated LZ or DZ cells. What are the top DGEs in these clusters? These might provide important insights into GC response in mLNs compared to pLNs.
4. Authors conclude "Collectively, these data show that signatures associated with positive selection and control of cell cycle progression are suppressed by Ga13 signaling in the mLN GC." This could be somehow misleading. From what I read in Fig. S4C-E, the comparison between Ga13 deficient vs sufficient also indicates a difference in pLNs. Please quantify the difference by paired comparison between Ga13 deficiency vs sufficiency and between pLNs vs mLNs. It is not wrong to focus on the comparison between Ga13 deficient mLN vs the rest as the authors did. However, the data appears to show that signatures associated with positive selection and control of cell cycle progression are suppressed by Ga13 signaling in both mLN and pLN GC. In addition, signatures associated with positive selection and control of cell cycle progression are stronger in mLN as compared to pLN GC, regardless of Ga13 expression. This distinction should be clearly stated.
5. The conclusion that PI3K/AKT plays a minor role in Ga13 regulation partially arises from the data showing the influence of PI3K/AKT on LZ/DZ polarisation, which was not observed in Ga13 deficiency. This could be more convincingly demonstrated with an experiment with KO AKT in the U46619 system, similar to those in Fig. 4G.

Minor questions.

6. Although the study focuses on positive selection to retain GC B cells, another outcome of GC selection is the formation of antibody-secreting cells (ASCs). Authors show some data on the memory cells but whether ASC compartment is influenced remains unknown. At least from the scRNA-seq, authors may identify pre-ASC populations with higher IRF4, Prdm1 and lower Bcl6.
7. Quantification is needed for Western blot (WB) blots (Fig. 4F, 5, and 7A) to support the conclusions drawn. For example, the authors state that "In the absence of glutamine, the ability of U46619 to reduce MYC expression was blunted". This is not entirely true. Without Gln, Myc expression was also reduced by the U46619 treatment.
8. The data in Fig. 7E is confusing, with Ga13-deficient GC showing both competitive advantages and disadvantages in different panels. The left panel shows competitive advantages, aligning with the results in Fig. 1F and the major conclusion of the paper, but Ga13 deficient GC were lower than the control in the right panel. Clarification is needed.

Author Rebuttal letter:

We thank the reviewers for their thoughtful review of our manuscript. Their insightful comments significantly improved our revised draft. Please find our detailed point-by-point responses below.

Reviewer #1

(Remarks to the Author)

In this manuscript, Nguyen et al. show that loss of Ga13 drives B cell lymphomas preferentially in the mesenteric lymph nodes, and argue that the mesenteric lymph nodes are unique because Ga13 is required to counteract mTORC1 signaling driven by the availability of dietary glutamine. Their study contributes to the very interesting questions of (1) why immune responses differ in differently situated lymph nodes, and (2) what is distinct about the gut microenvironment beyond the microbiome. The experiments are logical and well-controlled, and my comments are relatively minor.

We thank the reviewer for their positive comments.

- 1) The authors show that strong induction of pAKT by over-expression of myrAkt or loss

of PTEN does not phenocopy loss of Ga13 in mLN, although induction of pAKT may explain the effect of Ga13 loss in pLN. For completeness, it would be interesting to assess (for example by flow cytometry) whether pAKT is measurably changed in Ga13-KO GC B cells compared to control in the pLN or mLN.

The reviewer raises a good point.

Therefore, we assessed pAkt in GCB from pLN and mLN from $\text{G}\ddot{\text{i}}\text{j}13$ -deficient mixed chimeras by flow cytometry using methods previously described by Vitoria and colleagues (PMID: 28636954) that minimize artifacts potentially introduced during sample processing. We now show Fig. S3J-L $\hat{=}$ Flow cytometry for pAkt in mLN or pLN LZ GCB from control or $\text{G}\ddot{\text{i}}\text{j}13$ -deficient mixed chimeras. that pAkt is increased in $\text{G}\ddot{\text{i}}\text{j}13$ -deficient pLN LZ GC B cells but not mLN in mixed chimeras (Fig. S3J-L). These data support the hypothesis that dysregulation of Akt as a result of loss of $\text{G}\ddot{\text{i}}\text{j}13$ is not the principal driver of expansion of $\text{G}\ddot{\text{i}}\text{j}13$ -deficient GC B cells in the mLN.

2) The authors argue that the availability of glutamine in the mLN \rightarrow mTORC1 activation (upon loss of Ga13) \rightarrow Myc expression \rightarrow outgrowth. The evidence for mTORC1 activation in the mLN is in Fig. 3J,K, where the difference in pRPS6 upon loss of Ga13 is subtle. It would be helpful to extend this. Would flow cytometry enable analysis of a larger number of cells? What happens in the pLN? Does Q limitation affect pRPS6? Does Q supplementation activate the pathway (here it would be nice to see Myc and BrdU also)?

We thank the reviewer for raising these issues. We assessed pRPS6 in GCB from pLN and mLN from $\text{G}\ddot{\text{i}}\text{j}13$ -deficient mixed chimeras by flow cytometry again using methods previously described by Vitoria to minimize artifacts introduced during sample processing. We found that pRPS6 was increased in Fig. 3J-L $\hat{=}$ Flow cytometry for pRPS6 in mLN or pLN $\text{G}\ddot{\text{i}}\text{j}13$ -deficient LZ GC B cells in mLN and was LZ GCB from control or $\text{G}\ddot{\text{i}}\text{j}13$ -deficient mixed decreased in pLN (Fig. 3J-L). chimeras.

In mixed chimeras given glutamine/glutamic acid-deficient diet (-Q/E), pRPS6 was decreased in $\text{G}\ddot{\text{i}}\text{j}13$ -deficient mLN LZ GCB (Fig. 7D).

Fig. 7D $\hat{=}$ Ratio of gMFI of pRPS6 of $\text{G}\ddot{\text{i}}\text{j}13$ KO LZ GCB divided by WT (CD45.1/2) LZ GCB in mLN of mixed chimeras treated with control or glutamine/glutamic acid deficient (-Q/E) diet.

Regarding whether glutamine supplementation activates the pathway in vivo, we saw a trend toward increased Myc in $\text{G}\ddot{\text{i}}\text{j}13$ -deficient mLN GCB in mixed chimeras treated with glutamine water for 3 weeks but there was considerable variability in the data.

Therefore, to address whether glutamine supplementation had the capacity to activate the pathway, we fasted mixed chimeras for 24 hours with or without glutamine containing water and assessed the difference in Myc expression and proliferation between $\text{G}\ddot{\text{i}}\text{j}13$ -deficient and WT GC B cells. Administration of glutamine

water in fasted mice was able to restore the increase in Myc

Reviewer Figure 1 (A) $G\ddot{i}13$ -deficient mixed chimeras were fasted of food for expression and proliferation in $G\ddot{i}13$ - 24 hours with or without glutamine supplemented water. (B-C) Difference in deficient mLN GC B cells to levels Myc⁺ cells between $G\ddot{i}13$ -deficient and WT LZ GCB (B) or difference in BrdU⁺ cells between $G\ddot{i}13$ -deficient and WT GCB (C) in mLN of mixed chimeras that similar to fed mice (Reviewer Figure were fed, fasted of food for 24 hours or fasted of food with glutamine in the 1), drinking water for 24 hours (Fasted +Q).

3) In Figs. 6C, 6D, and 7D, it would be helpful to see the ratio of % Myc⁺ (or % BrdU⁺) among Ga13-KO GC B cells to the % Myc⁺ (or % BrdU⁺) among control GC B cells, from mice tx with vehicle and mice tx with rapamycin or dietary restriction. This would help distinguish the global effect of treatment on all GC B cells from effects specific to Ga13-KO GC B cells.

We agree with the reviewer. We now show the difference in Myc or BrdU positivity between $G\ddot{i}13$ KO and WT (CD45.1/2) GC B cells in 6C, 6D and 7D to show the specific effects on Myc expression and proliferation that occur in $G\ddot{i}13$ -deficient GC B cells as a result of rapamycin or dietary restriction of glutamine and glutamic acid.

Fig. 6C and 6D (A) Difference in Myc⁺ cells between $G\ddot{i}13$ KO LZ GCB and WT (CD45.1/2) LZ GCB or difference in BrdU⁺ cells between $G\ddot{i}13$ KO GCB and WT (CD45.1/2) GCB in mLN of mixed

Fig. 7D (A) Difference in Myc⁺ cells between $G\ddot{i}13$ chimeras treated with rapamycin. KO LZ GCB and WT (CD45.1/2) LZ GCB or difference in BrdU⁺ cells between $G\ddot{i}13$ KO GCB and WT (CD45.1/2) GCB in mLN of mixed chimeras treated with control or glutamine/glutamic acid deficient (-Q/E) diet.

Additionally, to distinguish global versus KO and tissue specific effects of perturbations in this pathway, we now include data showing that $G\ddot{i}13$ -deficient mLN GCB, but not pLN or PP GCB, are specifically dependent on the expression of the glutamine transporter Slc38a1/SNAT1 (Fig. 7E).

Fig. 7E (A) Ratio of sgRNA-expressing Cas9⁺ GCB (sgRNA⁺ GCB) divided by sgRNA⁺ FoB in mLN, pLN or PP of Aid-Cas9 or Aid-Cas9 Gna13f/f BM chimeras transduced with retrovirus targeting Slc38a1.

Reviewer #2

(Remarks to the Author)

In this manuscript Nguyen et al. study the role of Galpha13 in mucosal germinal centers (GC). Galpha13 is known to limit GC expansion by inhibiting Akt signaling. Here the authors show that Galpha13 deficiency increases the frequency of cMyc⁺ cells and mTORC1 activity. Moreover, the authors demonstrate that Galpha13 deficient GC B cells remain more competitive in the absence of gut microbiota or T cell help. Interestingly, Galpha13 deficient GC B cells seem to rely less on fatty acid metabolism than normal GC B cells and instead their expansion is (partially) supported by glutamine. In my opinion the suggested metabolic reprogramming in the absence of Galpha13 is the most interesting part of the study and could play a role in developing new therapies. Unfortunately, one the most interesting experiments (Fig.7F) is underpowered. This experiment should be expanded to include more experimental animals, wildtype controls and mice fed with other nutrients (for example glucose). This would allow to test whether a general increase in high-energy nutrients fastens lymphoma development in this mouse model or whether glutamine plays a special role.

We thank the reviewer for their insightful comments. We have now performed the glutamine supplementation experiment on additional cohorts of $G\ddot{i}13$ -deficient (Fig. 7G) and in $G\ddot{i}13$ -sufficient (Fig. S7I) animals.

4 of 13 $G\ddot{i}13$ -deficient animals

that received glutamine supplementation developed enlarged mLN with 25-fold or greater expansion of GC B cells (Fig. 7G). In contrast, no glutamine treated $\text{G}\ddot{\text{i}}\text{j}13$ -sufficient or control $\text{G}\ddot{\text{i}}\text{j}13$ -deficient animals developed Fig. 7G - Long-term glutamine supplementation in $\text{G}\ddot{\text{i}}\text{j}13$ -deficient mice.

mesenteric lymphadenopathy with GC B cell expansions at this timepoint.

While long-term in vivo experiments with supplementation of other nutrients such as glucose is feasible, our current data do not support a role for glucose in our model of lymphoma. Glucose and glycolysis do not differentially support expansion of Fig. S7I - Long-term glutamine supplementation in WT mice.

$\text{G}\ddot{\text{i}}\text{j}13$ -deficient lymphoma cells in vitro (Fig. S6H and S6I). Additionally, glucose transport via $\text{Slc}2\text{a}1/\text{Glut}1$ (Fig. 6L) does not differentially support $\text{G}\ddot{\text{i}}\text{j}13$ -deficient mLN GC B cells in vivo.

Regarding whether glutamine plays a special role in supporting $\text{G}\ddot{\text{i}}\text{j}13$ -deficient GC B cells in the mLN, we now provide data showing that $\text{G}\ddot{\text{i}}\text{j}13$ -deficient mLN GCB are Fig. 7E - Ratio of sgRNA-expressing Cas9+ GCB (sgRNA+ GCB) divided specifically dependent on the by sgRNA+ FoB in mLN, pLN or PP of Aid-Cas9 or Aid-Cas9 $\text{Gna}13\text{f}/\text{f}$ BM chimeras transduced with retrovirus targeting $\text{Slc}38\text{a}1$. expression of the glutamine transporter $\text{Slc}38\text{a}1/\text{SNAT}1$ (Fig. 7E).
Minor concerns:

- Throughout the manuscript the authors show the percentage of GC B cells in the population of life cells. The authors should add a graph showing the percentage of GC B cells in the population of total B cells.

We thank the reviewer for raising this issue. We have now included additional panels in the supplemental data showing the percentage of GC B cells in the population of total B cells. These graphs are now included in Figures S2A, S6A, S6C, S6D, S6E and S7H.

- In Fig.1K the authors should specify the markers used to define memory B cells in the figure legends

We thank the reviewer for pointing this out. Memory B cells are defined as $\text{B}220+\text{IgD}^{\text{lo}}\text{CD}38^{\text{high}}\text{Fas}^{\text{int}}\text{GL}7^{-}$ as shown in the example flow cytometry plot in 1H. This information has been added to the legend for 1K.

- The authors should specify the genotype of controls used. Where the control mice cre positive? If not, can the authors provide evidence to exclude the contribution of cre recombinase expression to the observed phenotype.

We thank the reviewer for raising this issue. For S1PR2-creERT2 (Fig. 1H-K) and Aid-Cas9 (Fig. 2D-F) experiments all control mice expressed Cre. While the genotypes for the control mice was described in most of the figures, we apologize for not including this information for Fig. 1D, 6I and 6J. Some of the control mice in these experiments expressed Cre and we have amended the legend for these figures accordingly. For mixed BM chimera experiments, control BM was from littermate animals that were $\text{Gna}13\text{f}/+$. We have used Cr2-cre in mixed bone chimeras extensively in our previous studies (PMID: 31506281, 33237303 and 35759728) and have not observed the constellation of findings associated with loss of $\text{Gna}13$ in this manuscript in any of those other systems. Therefore, we have a high degree of confidence that the phenotypes reported here are not a result of Cre expression in GC B cells.

- The authors write "Galpha13 suppresses mLN GC proliferation independently of PI3K/Akt" and "in the absence of Galpha13, loss of Pten promoted GC outgrowths and LZ expansion suggesting that Pten and Galpha13 suppress GC B cell expansion via distinct pathways". These statements are in my opinion unjustified. The presented data do not exclude that Galpha13 suppresses mLN GC proliferation by activating PI3K/Akt. One could imagine different scenarios in which Galpha13 deletion does not lead to LZ polarization, despite Akt signaling driving GC expansion. For example, it is possible that while Galpha13 loss, Pten loss as well as Akt-hyperactivation all lead to an increase in signaling downstream of Akt, the magnitude of signaling activity may differ. An intermediate increase in Akt signaling in Galpha13-deficient B cells might support survival, but not polarization unlike a stronger Akt signal in Pten deficient or Myr-Akt expressing B cells. The authors should assess activity of the Akt signaling pathway in their models and change the language of the text to avoid misleading statements. The reviewer makes an excellent point. We agree that we cannot exclude transient or subtle changes in Akt in vivo as contributing to the phenotype in Gij13-deficient GCB in mLN, we have amended our language to be more precise. Importantly, we now show by flow cytometry that pAkt is increased in Gij13-deficient pLN LZ GC B cells but not mLN in mixed chimeras (Fig. S3J-L).

We also now include additional analyses of our scRNAseq dataset showing that the hallmark gene signature PI3K_AKT_MTOR_SIGNALING is down-regulated in KO mLN compared to all other samples (Fig. S4H) or when compared individually to WT mLN or KO pLN (Fig. 3D). Fig. S3J-L - Flow cytometry for pAkt in mLN or pLN LZ GCB from control or Gij13-deficient mixed chimeras.

Fig. S4H - Gene set enrichment plot for PI3K_AKT_MTOR_SIGNALING of Gij13-deficient mLN GCB compared to all other samples.

- In Fig.2K the authors should show an example staining for pS6

We believe the reviewer is referring to the data in Fig. 3K, which is quantification of imaging data an example of which is shown in 3J. The workflow for quantification of pRPS6 from imaging data is described in the methods.

We now also include FACS data showing that pRPS6 is increased in Gij13-deficient mLN GCB but not pLN GCB in mixed chimeras (Fig. 3J-L). Fig. 3J-L - Flow cytometry for pRPS6 in mLN or pLN LZ GCB from control or Gij13-deficient mixed chimeras.

L).

- The authors write "We found that in the presence of silvestrol and MLN4924, Galpha13 stimulation was not able to further reduce MYC protein (Fig. 5E). Collectively, these data suggest that sustained Galpha13-mediated loss of MYC protein is primarily a result of reduced translation (Fig. 5F)". I think this statement cannot be made. In the shown experiment (Fig.5E) Myc protein levels are already very low when treated with the combination of MLN4924 and silvestrol, the band after U46619 treatment appears to be slightly weaker. The authors might not be able to observe strong effects of U46619 treatment because Myc levels before the treatment are already very low. I don't think this experiment convincingly shows that Galpha13-mediated loss of MYC protein is primarily a result of reduced translation. The reviewer raises a good point. Gij13 stimulation reduces MYC protein expression when proteasomal-dependent is blocked by MLN4924 (Fig. 5D). This degree of MYC

protein loss is out of proportion to the reduction in MYC mRNA (Fig. 5A). Therefore, we feel that the most likely explanation is that G β 13-mediated loss of MYC protein is a result of reduced translation. Nevertheless, we have amended our language to be more cautious in our interpretation of the data with respect to non-transcriptional, degradation-independent loss of MYC protein expression that occurs following G β 13-signaling.

Reviewer #3

(Remarks to the Author)

Muppidi JR and colleagues have previously shown that Ga13 deficiency in mature B cells leads to an increase in GC B cells in mLNs and the development of GCB-DLBCL. This study investigates the selective impact of Ga13 deficiency in mLNs over pLNs. Through genetic tools, scRNA-seq, and metabolic assays, the authors suggest that tumorigenesis from Ga13 deficiency primarily involves the mTORC1/Myc pathway, rather than PI3K/AKT. They also present evidence suggesting that dietary glutamine plays a significant role in regulating GC dynamics in mLNs in the context of Ga13 deficiency. The novelty of this study lies in its mechanistic insights and the proposed role of dietary metabolites, specifically glutamine, in differentially influencing oncogenic pathways in mLNs and pLNs, provided the findings are strongly supported.

We thank the reviewer for their thoughtful review of our manuscript.

Major questions to be addressed

1. The pivotal role of glutamine in preferential tumorigenesis due to Ga13 deficiency in mLNs versus pLNs is unclear. The authors show data on Gln supplementation in drinking water causing lymphadenopathy in mLNs (Fig. 7F), but the study has limitations, including a small sample size and the absence of a control group with Ga13 sufficient mice also receiving Gln supplementation.

The reviewer raises a good point. We performed the glutamine supplementation experiment on additional cohorts of G β 13-deficient (Fig. 7G) and G β 13-sufficient (Fig. S7I) animals.

4 of 13 G β 13-deficient animals that received glutamine supplementation developed enlarged mLN with 25-fold or greater expansion of GC B cells (Fig. 7G). In contrast, no glutamine treated G β 13-sufficient or control G β 13-deficient animals developed mesenteric lymphadenopathy or GC B cell expansions at this timepoint.

Fig. 7G - Long-term glutamine supplementation in G β 13-deficient mice. Fig. S7I - Long-term glutamine supplementation in WT mice.

2. The study does not confirm whether the combination of Ga13 deficiency and Gln supplementation is enough to induce DLBCL in pLNs, as it does in mLNs. A possible approach would be to increase Gln levels in pLNs through systemic supplementation and observe any increase in lymphadenopathy, specifically under Ga13 deficiency, as a sign of DLBCL development.

The reviewer raises an interesting idea. We, however, did not observe any lymphadenopathy in pLN of G β 13-deficient animals placed on long-term glutamine nor did we observe that outgrowths of G β 13-deficient GC B cells in peripheral lymph nodes of mixed chimeras could be induced with dietary glutamine supplementation for shorter periods. We speculate that this is related to anatomic differences and the quantities of dietary nutrients delivered to mLN compared to pLN. In mLN, lymph from large portions of the small intestine is delivered to each segment of the mLN. G β 13-deficient GC B cells have greater access to areas near lymphatics such as the follicular mantle and thus may be more influenced by higher local concentrations of dietary factors delivered via lymphatics. In contrast, in the peripheral lymph node, diet derived nutrients would be delivered via blood to interfollicular areas and would not be as concentrated as they would be in gut-draining lymphatics delivered to the mLN. Therefore, we would not necessarily expect that dietary supplementation of glutamine would induce outgrowths or lymphomas of G β 13-deficient GC B cells in pLN.

3. scRNA-seq provides valuable information for the changes caused by Ga13 deficiency, i.e. clusters 3, 4 and 15 expanded in mLNs and further enhanced by Ga13 deficiency. However, it is somehow disappointing that these clusters have not been thoroughly characterized. They were not annotated, though being excluded from

annotated LZ or DZ cells. What are the top DGEs in these clusters? These might provide important insights into GC response in mLNs compared to pLNs.

We agree with the reviewer that additional characterization of these clusters is needed. We have now included data showing the top differentially expressed genes in Clusters 3, 4 and 15 (Fig. S4F). Clusters 3 and 4 show a marked enrichment for expression of ribosomal genes which is consistent with the hallmark gene sets that are enriched in these clusters.

Fig. S4F â Differential expressed genes in clusters 3, 4 and 15 compared to all other clusters.

4. Authors conclude âCollectively, these data show that signatures associated with positive selection and control of cell cycle progression are suppressed by Ga13 signaling in the mLN GC.â This could be somehow misleading. From what I read in Fig. S4C-E, the comparison between Ga13 deficient vs sufficient also indicates a difference in pLNs. Please quantify the difference by paired comparison between Ga13 deficiency vs sufficiency and between pLNs vs mLNs. It is not wrong to focus on the comparison between Ga13 deficient mLN vs the rest as the authors did. However, the data appears to show that signatures associated with positive selection and control of cell cycle progression are suppressed by Ga13 signaling in both mLN and pLN GC. In addition, signatures associated with positive selection and control of cell cycle progression are stronger in mLN as compared to pLN GC, regardless of Ga13 expression. This distinction should be clearly stated.

The reviewer raises an excellent point. We now include comparisons of enriched gene signatures comparing Gna13 KO mLN to all others combined and as well as individual samples (Fig. 3D). Additionally, we now include comparisons of WT mLN to WT pLN and Gna13 KO pLN compared to all others and to WT pLN (Fig. S4G). As the reviewer correctly states, there is an increase in many of these gene signatures when comparing KO pLN to WT pLN or WT mLN to WT pLN. However, the degree of enrichment of signatures associated with Myc and mTorc1 amongst the four samples is highest in the Gna13 KO mLN sample when compared to all others or to the individual samples. We have added language clarifying this point in Fig. 3D â Gene set enrichment analysis of Gij13-deficient mLN GCB compared to all other samples the revised manuscript. combined or individually.

Fig. S4G â Gene set enrichment analysis of WT mLN compared to pLN or Gij13-deficient pLN GCB compared to all other samples combined or to WT pLN.

5. The conclusion that PI3K/AKT plays a minor role in Ga13 regulation partially arises from the data showing the influence of PI3K/AKT on LZ/DZ polarisation, which was not observed in Ga13 deficiency. This could be more convincingly demonstrated with an experiment with KO AKT in the U46619 system, similar to those in Fig. 4G.

We thank the reviewer for raising this question. In our CRISPR screen data, the guide abundance for AKT1, AKT2, AKT3 or PTEN were not altered in the presence of Gij13 stimulation with U46619 in contrast to guides for GNA13, ARHGEF1 and RIC8A which markedly increased with U46619 (Reviewer Fig. 1A and B).

Additionally, we targeted AKT1, AKT2 and AKT3 in NUDUL1 and OCI-Ly8 and PTEN in NUDUL1. Reviewer Figure 2 (A-B) Crispr screen scores (CSS) in Day 21 U46619 treated cells and found that, in the absence of versus Day 0 (y-axis) compared to Day 21 untreated cells versus Day 0 (x-axis) in single AKT isoforms or PTEN, NUDUL1 (A) or OCI-Ly8 (B). (C-E) Phospho-P70S6K T389, phospho-Akt S473 and MYC expression in U46619-treated NUDUL1 or OCI-Ly8 cells expressing sgRNA U46619 was still able to suppress targeting AKT1, AKT2, AKT3 or PTEN. mTORC1 activity and MYC expression (Reviewer Fig. 2C-E).

Minor questions.

6. Although the study focuses on positive selection to retain GC B cells, another outcome of GC selection is the formation of antibody-secreting cells (ASCs). Authors show some data on the memory cells but whether ASC compartment is influenced remains unknown. At least from the scRNA-seq, authors may identify pre-ASC populations with higher IRF4, Prdm1 and lower Bcl6.

The reviewer raises an excellent point. We took 2 approaches to answer this question. First, we analyzed the frequency of reporter positive plasma cells (PCs) in the bone marrow (BM) of S1pr2-tdTomato Gna13f/f or control (S1pr2-tdTomato Gna13f/+) mice at 8 weeks following tamoxifen administration. We found that loss of G β 13 did not increase the frequency of reporter positive PCs in the bone marrow (Fig. S2G).

Because BM PCs are derived from diverse sources, it was not clear whether there was increased PC production specifically from the mLN. Therefore, we also

+

Fig. S2G- tdTomato Bone marrow plasma cells in S1pr2-tdTomato f/+ f/f Gna13 or Gna13 mice at 8 weeks following tamoxifen administration. assessed the frequency pre-PC

populations in our scRNAseq dataset from mLN and pLN. In the data shown in Figures 3 and S4, we excluded Bcl6^{lo} populations from our gene expression analyses and therefore pre-PC population frequencies cannot be assessed from clustering shown in the manuscript. When Bcl6^{lo} populations are included in the analysis, 2 clusters of Irf4^{hi} Prdm1^{hi} Bcl6^{lo} plasmablasts can be identified (PB1 and PB2; Reviewer Fig. 3). PB1 and PB2 cells are not increased in the G β 13 KO mLN in comparison to WT mLN (Reviewer Fig. 3). Collectively, these data suggest that loss of G β 13 does not increase PC

Reviewer Figure 3 Frequency of plasmablasts (PB1 and PB2; Bcl6^{lo} differentiation from the mLN GC. Prdm1^{hi} IRF4^{hi}) in WT or G β 13-deficient pLN or mLN.

7. Quantification is needed for Western blot (WB) blots (Fig. 4F, 5, and 7A) to support the conclusions drawn. For example, the authors state that in the absence of glutamine, the ability of U46619 to reduce MYC expression was blunted. This is not entirely true. Without Gln, Myc expression was also reduced by the U46619 treatment.

We thank the reviewer for pointing this out. We have now included quantification of our western blot experiments and clarified our wording. In 7A, there is a reduction of Myc induced by U46619 in the absence of glutamine but this reduction is not as large as the reduction of Myc induced by U46619 in the presence of glutamine. To provide a more quantitative assessment of this point by FACS, we show that the loss of Myc induced by G β 13 signaling in the absence of glutamine is not as strong as when glutamine is

present for the duration of cell culture (Fig. S7A). Importantly, in the presence of U46619, addition of glutamine to glutamine-starved cells for 1 hour prior to analysis was no longer able to restore Myc expression.

Figure S7A MYC MFI in NUDUL1 cells in the presence or absence of U46619 that were also cultured in the absence of glutamine for 4 h or in the absence of glutamine for 3 hours with glutamine added back for 1 hour prior to analysis.

8. The data in Fig. 7E is confusing, with Ga13-deficient GC showing both competitive advantages and disadvantages in different panels. The left panel shows competitive advantages, aligning with the results in Fig. 1F and the major conclusion of the paper, but Ga13 deficient GC were lower than the control in the right panel. Clarification is needed.

We apologize for the confusion. The BM mixing ratio for this experiment is 15% $\text{G}\ddot{\text{i}}13$ -deficient CD45.2 and 85% CD45.1/2. The frequency of CD45.2 $\text{G}\ddot{\text{i}}13$ -deficient cells in the naïve FoB compartment is on average ~15%. The frequency of CD45.2 $\text{G}\ddot{\text{i}}13$ -deficient GC B cells amongst GC B cells in the control group in this experiment is 40.57% on average and goes up to an average of 53% in the glutamine treated group (Reviewer Figure 4). The reason for the apparent higher percentage of WT (CD45.1/2) GC B cells as a percentage of live cells in the control group is because high starting frequency (85%) of the WT BM. This experiment shows a cell-intrinsic competitive advantage for $\text{G}\ddot{\text{i}}13$ -deficient GC B cells in the mLN that is increased in the presence of glutamine supplementation in the diet.

Reviewer Figure 4 Percentages of CD45.2 follicular B cells (FoB) and GC B cells (GCB) in mesenteric LN (mLN) of $\text{G}\ddot{\text{i}}13$ -deficient mixed BM chimeras that were treated with glutamine (+Q) in drinking water for 3 weeks prior to analysis.

Version 1:

Decision Letter:

Our ref: NI-A37208A

3rd Jun 2024

Dear Jagan,

Thank you for submitting your revised manuscript " $\text{G}\alpha13$ restricts nutrient driven proliferation in mucosal germinal centers" (NI-A37208A). It has now been seen by the original referees and their comments are below. The reviewers find that the paper has improved in revision, and therefore we'll be happy in principle to publish it in Nature Immunology, pending minor revisions to satisfy the referees' final requests and to comply with our editorial and formatting guidelines.

We will now perform detailed checks on your paper and will send you a checklist detailing our editorial and formatting requirements in about a week. Please do not upload the final materials and make any revisions until you receive this additional information from us.

If you had not uploaded a Word file for the current version of the manuscript, we will need one before beginning the editing process; please email that to immunology@us.nature.com at your earliest convenience.

Thank you again for your interest in Nature Immunology Please do not hesitate to contact me if you have any questions.

Kind regards,

Laurie

Laurie A. Dempsey, Ph.D.
Senior Editor
Nature Immunology
l.dempsey@us.nature.com
ORCID: 0000-0002-3304-796X

Reviewer #1 (Remarks to the Author):

In this manuscript, Nguyen et al. show that loss of Ga13 drives B cell lymphomas preferentially in the mesenteric lymph nodes, because Ga13 limits mTORC1 signaling driven by the availability of dietary glutamine. The authors have comprehensively addressed my concerns, and the revised manuscript will be of interest for its contributions to our understanding of how immune responses differ among lymph nodes and to our understanding of cancer biology.

Reviewer #2 (Remarks to the Author):

Thank you for addressing my comments to the manuscript.
All of my concerns have been answered in a satisfactory manner and I have no further suggestions.

Reviewer #3 (Remarks to the Author):

The authors have presented several new experiments addressing my questions. These results convincingly demonstrate that Gln supplementation selectively promotes lymphadenopathy in mLNs in the context of Ga13 deficiency, with supporting evidence indicating the dependence on the Gln transporter Slc38a1. The findings are further substantiated by the characterization of scRNA-seq data and plasma cell differentiation in Ga13-deficient mice.

I have no further questions but a minor suggestion. It would be valuable to include the following results, as stated by the authors in their response, in the supplementary materials:

"We did not observe any lymphadenopathy in the pLN of Ga13-deficient animals placed on long-term glutamine, nor did we observe outgrowths of Ga13-deficient GC B cells in peripheral lymph nodes of mixed chimeras induced by dietary glutamine supplementation for shorter periods."

Including these results would provide an important control, highlighting the tissue-specific regulation likely due to the gut lymphatics, as suggested by the authors.

Version 2:

Decision Letter:

In reply please quote: NI-A37208B

Dear Dr. Muppidi,

I am delighted to accept your manuscript entitled "Ga13 restricts nutrient driven proliferation in mucosal germinal centers" for publication in an upcoming issue of Nature Immunology.

Over the next few weeks, your paper will be copyedited to ensure that it conforms to Nature Immunology style. Once your paper is typeset, you will receive an email with a link to choose the appropriate publishing options for your paper and our Author Services team will be in touch regarding any additional information that may be required.

Please note that *Nature Immunology* is a Transformative Journal (TJ). Authors may publish their research with us through the traditional subscription access route or make their paper immediately open access through payment of an article-processing

charge (APC). Authors will not be required to make a final decision about access to their article until it has been accepted. Find out more about Transformational Journals.

Authors may need to take specific actions to achieve compliance with funder and institutional open access mandates. If your research is supported by a funder that requires immediate open access (e.g. according to Plan S principles) then you should select the gold OA route, and we will direct you to the compliant route where possible. For authors selecting the subscription publication route, the journal's standard licensing terms will need to be accepted, including self-archiving policies. Those licensing terms will supersede any other terms that the author or any third party may assert apply to any version of the manuscript.

Your paper will be published online soon after we receive your corrections and will appear in print in the next available issue.

Also, if you have any spectacular or outstanding figures or graphics associated with your manuscript - though not necessarily included with your submission - we'd be delighted to consider them as candidates for our cover. Simply send an electronic version (accompanied by a hard copy) to us with a possible cover caption enclosed.

If you have not already done so, we strongly recommend that you upload the step-by-step protocols used in this manuscript to protocols.io. protocols.io is an open online resource that allows researchers to share their detailed experimental know-how. All uploaded protocols are made freely available and are assigned DOIs for ease of citation. Protocols can be linked to any publications in which they are used and will be linked to from your article. You can also establish a dedicated workspace to collect all your lab Protocols. By uploading your Protocols to protocols.io, you are enabling researchers to more readily reproduce or adapt the methodology you use, as well as increasing the visibility of your protocols and papers. Upload your Protocols at <https://protocols.io>. Further information can be found at <https://www.protocols.io/help/publish-articles>.

Please note that we encourage the authors to self-archive their manuscript (the accepted version before copy editing) in their institutional repository, and in their funders' archives, six months after publication. Nature Portfolio recognizes the efforts of funding bodies to increase access of the research they fund, and strongly encourages authors to participate in such efforts. For information about our editorial policy, including license agreement and author copyright, please visit www.nature.com/ni/about/ed_policies/index.html

An online order form for reprints of your paper is available at https://www.nature.com/reprints/author-reprints.html. Please let your coauthors and your institutions' public affairs office know that they are also welcome to order reprints by this method.

Sincerely,

Jamie D K Wilson, D.Phil
Chief Editor
Fir:

Laurie A. Dempsey, Ph.D.
Senior Editor
Nature Immunology
l.dempsey@us.nature.com

ORCID: 0000-0002-3304-796X

Click here if you would like to recommend Nature Immunology to your librarian
<http://www.nature.com/subscriptions/recommend.html#forms>

** Visit the Springer Nature Editorial and Publishing website at http://editorial-jobs.springernature.com?utm_source=ejp_NImm_email&utm_medium=ejp_NImm_email&utm_campaign=ejp_NImm for more information about our career opportunities. If you have any questions please click [here](mailto:editorial.publishing.jobs@springernature.com).
